# Efficient Diversity-Preserving Diffusion Alignment via Gradient-Informed GFlowNets

**Zhen Liu**[1,2,3,†]   **Tim Z. Xiao**[2,4,*]   **Weiyang Liu**[2,5,*]   **Yoshua Bengio**[1]   **Dinghuai Zhang**[1,6,†]

[1]Mila, Université de Montréal   [2]Max Planck Institute for Intelligent Systems - Tübingen
[3]The Chinese University of Hong Kong (Shenzhen)   [4]University of Tübingen
[5]University of Cambridge   [6]Microsoft Research   [†]Corresponding author   [*]Equal contribution

Project page: **nabla-gfn.github.io**

## Abstract

While one commonly trains large diffusion models by collecting datasets on target downstream tasks, it is often desired to align and finetune pretrained diffusion models with some reward functions that are either designed by experts or learned from small-scale datasets. Existing post-training methods for reward finetuning of diffusion models typically suffer from lack of diversity in generated samples, lack of prior preservation, and/or slow convergence in finetuning. In response to this challenge, we take inspiration from recent successes in generative flow networks (GFlowNets) and propose a reinforcement learning method for diffusion model finetuning, dubbed Nabla-GFlowNet (abbreviated as $\nabla$-GFlowNet), that leverages the rich signal in reward gradients for probabilistic diffusion finetuning. We show that our proposed method achieves fast yet diversity- and prior-preserving finetuning of Stable Diffusion, a large-scale text-conditioned image diffusion model, on different realistic reward functions.

## 1 Introduction

Diffusion models [17, 61, 51] are a powerful class of generative models that model highly complex data distributions with sequential denoising steps. They prove capable of modeling distributions of images [51, 9, 49], videos [19, 18, 4], 3D objects [81, 47, 34, 35, 13], molecules [72, 20, 32], languages [37, 53, 36] and many others. State-of-the-arts diffusion models for downstream applications are typically large in network size and demand a significant amount of data to train.

It is often desirable that one finetunes pretrained diffusion models with some reward — either learned from human preferences as in reinforcement learning from human feedback (RLHF) [5, 43] or designed by human experts [58, 41]. While existing methods achieve fast reward convergence [71, 6], many of them are diversity-lacking, prior-ignoring, and/or computationally expensive.

To address this issue, we take the best from both traditional reinforcement learning (RL) and direct reward maximization approaches and propose a novel reinforcement learning objective, dubbed $\nabla$-DB, that leverages the rich information in reward gradients. Our method is rooted in both generative flow networks (GFlowNets) [2], a framework and language for generative modeling on a direct acyclic transition graph, and soft RL [15, 16]. Compared to existing gradient-informed methods, our probabilistic finetuning method achieves better sample diversity. We further propose $\nabla$-GFlowNet with the objective *residual* $\nabla$-DB that, by leveraging the structure of diffusion models, allows us to perform fast, diversity preserving and prior-preserving amortized finetuning of diffusion models with rather long sampling sequences.

We summarize our major contributions below:

- We propose $\nabla$-DB and the pretrained-model-aware variant *residual* $\nabla$-DB, the objectives that efficiently leverage the rich information in reward gradients in a principled and probabilistic way.

- Our $\nabla$-GFlowNet is the first GFlowNet method that considers first-order information in reward signals, and is closely connected to soft reinforcement learning.

- We empirically show that with the proposed *residual* $\nabla$-DB objective, we may achieve diversity- and prior-preserving yet fast finetuning and alignment of diffusion models.

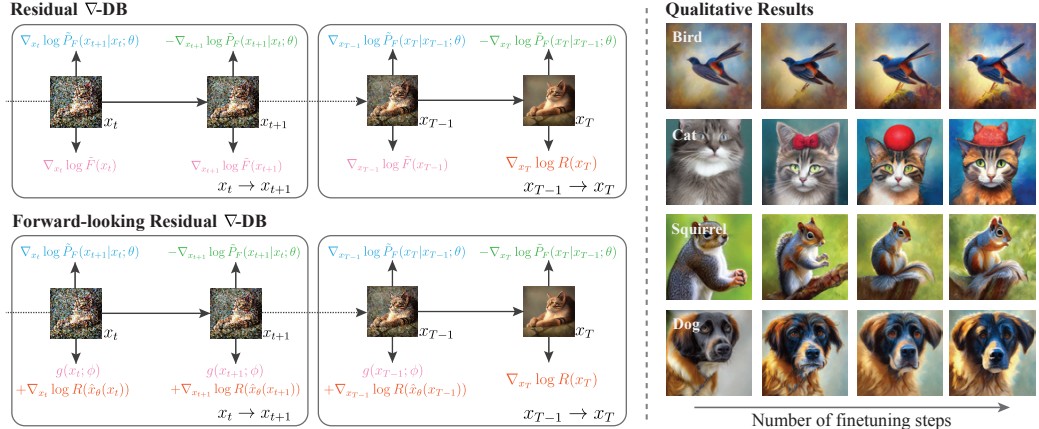

Figure 1: Left: Illustration of the proposed *residual* ∇-DB objective, along with its forward-looking variant. The two "forces" on each image in each transition $x_t \to x_{t+1}$ out of a trajectory $\tau = (x_0, x_1, ..., x_T)$ are expected to sum to zero. Green and blue terms represent forward and reverse residual policy scores (respectively), orange terms represent signals from terminal rewards and pink terms represent flow scores or residual flow scores, each of which is defined in Section 3. Notice that the reward term on the $x_T$ in the final transition are different from the others. Right: Generated image from a model finetuned with the proposed *residual* ∇-DB on the Aesthetic Score reward. The text prompt for each row is shown on the left. The leftmost figure is the image generated by the pretrained model while the rightmost one is from the model finetuned for 200 iterations.

## 2 PRELIMINARIES

### 2.1 DIFFUSION MODELS AND RL-BASED FINETUNING

Diffusion models [17, 59, 61] are a class of hierarchical latent models that model the generation process as a sequence of denoising steps. Different from the convention in diffusion model literature, for convenience we adopt in this paper the reverse time of arrow where $x_T$ means samples from the data distribution and the sampling process starts from $t = 0$. Under this convention, the probability of the generated samples is:

$$P_F(x_T) = \int_{x_{0:T-1}} P_0(x_0) \prod P_F(x_t|x_{t-1}) dx_{0:T-1}. \tag{1}$$

Here $P_0(x_0)$ is a fixed initial distribution, $P_F(x_T)$ is the likelihood of the model generating data $x_T$, and the noisy states $x_t$ in the intermediate time steps are constructed by a pre-defined noising process, and the *forward policy* $P_F(x_t|x_{t-1})$ is the denoising step of the diffusion model[1]. Take DDPM [17] as an example: the corresponding noising process is $q(x_{t-1}|x_t) = \mathcal{N}(\sqrt{\alpha_{t-1}/\alpha_t}x_t, \sqrt{1 - \alpha_{t-1}/\alpha_t}I)$ and induces $q(x_t|x_T) = \mathcal{N}(\sqrt{\alpha_{T-t}}x_T, \sqrt{1 - \alpha_{T-t}}I)$, where $\{\alpha_t\}_t$ is a noise schedule set. With this noising process defined, the training loss is:

$$\mathbb{E}_{t\sim\text{Uniform}(\{1,...,T\}),\epsilon\sim\mathcal{N}(0,I),x_T\sim\mathcal{D}} w(t)\left\|x_\theta(\sqrt{\alpha_{T-t}}x_T + \sqrt{1 - \alpha_{T-t}}\epsilon, t) - x_T\right\|^2, \tag{2}$$

where $\mathcal{D}$ is a dataset, $w(t)$ is a certain schedule weighting function, and $x_\theta(x_t, t)$ is a data prediction model that predict the clean data $x_T$ given a noisy data $x_t$ at time step $t$.

The sequential sampling process of diffusion models is Markovian and one could construct an Markov desicion process (MDP) to describe its denoising process. Similar to existing approaches [3, 80], given a Markovian diffusion sampling algorithm, we construct an MDP by treating the noisy sample $(x_t, t)$ at each diffusion inference step $t$ as a state and a denoising step from $x_t$ to $x_{t+1}$ as a transition. Given such an MDP defined, one may finetune diffusion models with techniques like DDPO [3] by collecting on-policy sample trajectories $\{(x_1, ..., x_T)\}$ and optimize the forward (denoising) policy with a given terminal reward $R(x_T)$. We give a more detailed overview of MDP construction and DDPO in Appendix F.

---

[1]To clarify, the "forward" and "backward" directions in the GFlowNet literature are typically the opposite of those in the diffusion model literature.

## 2.2 RL FINETUNING IN THE GFLOWNET FRAMEWORK

It is well known [15, 50] in the soft RL literature that, by maximizing $\mathbb{E}_{\pi_\theta} \sum_t [R(s_T) + \frac{1}{\beta} \mathcal{H}(\pi_\theta(\cdot|s_t))]$ in an MDP (with state $s_t$, terminating state $s_T$, action $a_t$, terminal reward $R(\cdot)$, discount rate $\gamma$, policy $\pi_\theta$ and entropy function $\mathcal{H}$), we have $P(x) \propto \exp(\beta R(x))$ for any terminating state $x$. Finding optimal policies to sample from $R(\cdot)$ can therefore be achieved with appropriate designs of MDPs. While we may derive our objective with the soft RL language, here we follow a recent work [80] and adopt the GFlowNet [2] framework for generative models that sample terminal states with respect to reward with MDP defined on a directly acyclic graph (DAG). The GFlowNet framework uniquely motivates the introduction of a special intermediate reward associated to a reference backward policy, which is not obvious in the soft RL framework. Furthermore, in modeling distributions with a DAG-based MDP, the GFlowNet framework is more general and perhaps more natural than soft RL. We discuss the connections between two frameworks in Appendix D.

In a GFlowNet, the forward sampling policy $P_F(s'|s)$ on a DAG from an initial state $s_0$ to a terminal state $s_f$ is paired with a backward "reference" policy $P_B(s|s')$. One can view $P_F$ as distributing probability mass from $s$ to its successors, and $P_B$ as moving mass backward from $s'$ to its ancestors. When $P_F$ and $P_B$ are matched, they induce an unnormalized density $F(s)$ at each state, called the flow function. One way to specify the matching between $P_F$ and $P_B$ is through the following:

**Detailed Balance (DB).** A valid GFlowNet with a forward policy $P_F(s'|s)$, a backward policy $P_B(s|s')$, and a flow function $F(s)$ satisfies the following DB condition for all transition $(s \to s')$

$$P_F(s'|s)F(s) = P_B(s|s')F(s'). \tag{3}$$

Hence we have the following GFlowNet DB loss in the logarithm probability space:

$$L_{\text{DB}}(s, s') = \Big( \log P_F(s'|s) + \log F(s) - \log P_B(s|s') - \log F(s') \Big)^2 \tag{4}$$

with an extra terminal constraint $F(s_f) = R(s_f)$ to incorporate the target reward information.

In the context of time-indexed sampling processes such as diffusion models, the transition graph of states $s \triangleq (x_t, t)$ is naturally acyclic, as it adheres to the arrow of time [75]. With slight abuse of terminology that we use $x_t$ to represents the tuple $(x_t, t)$ when necessary, for time-indexed settings the *forward policy* is $P_F(x_{t+1}|x_t)$, the *backward policy* is $P_B(x_t|x_{t+1})$, and the *flow function* is $F(x_t)^2$. The corresponding DB condition is therefore

$$P_F(x_{t+1}|x_t)F(x_t) = P_B(x_t|x_{t+1})F(x_{t+1}). \tag{5}$$

To finetune a diffusion model with DB losses [80], one can simply set $P_F(x_{t+1}|x_t)$ to be the sampling process and fix $P_B(x_t|x_{t+1})$ to be the noising process used by the pretrained diffusion model. If $P_B$ is set to be fixed, we can similarly derive this condition using soft Q-learning [15], with details provided in Appendix D.

# 3 REWARD FINETUNING VIA GRADIENT-INFORMED GFLOWNETS

## 3.1 $\nabla$-DB: THE GRADIENT-INFORMED DETAILED BALANCE

In our setting, we do not have access to any dataset of images, but are given an external positive-valued reward function $R(\cdot)$ to which a generative model is trained to adapt. While a typical GFlowNet-based algorithm can effectively achieve this objective with diversity in generated samples, it only leverages the zeroth-order reward information and does not leverage any differentiability of the reward function. Yet, whenever the reward gradients are available, it is often beneficial to incorporate them into the finetuning objective since they help navigate the finetuned model on the optimization landscape and may significantly accelerate the optimization process. We are therefore motivated to develop $\nabla$-GFlowNet, a method that builds upon GFlowNet-based algorithms to take full advantage of the reward gradient signal. To achieve this, we take derivatives on the logarithms of both sides of the DB condition (logarithm of Equation 5) with respect to $x_{t+1}$ and obtain a necessary condition, which we call the forward[3] $\nabla$-DB condition:

$$\nabla_{x_{t+1}} \log P_F(x_{t+1}|x_t) = \nabla_{x_{t+1}} \log P_B(x_t|x_{t+1}) + \nabla_{x_{t+1}} \log F(x_{t+1}), \tag{6}$$

---

[2]We write $F_t(x_t)$ as $F(x_t)$ for the sake of notation simplicity.

[3]This is because the derivative is taken with respect to $x_{t+1}$.

and hence the corresponding forward $\nabla$-DB objective $L_{\overrightarrow{\nabla}\text{DB}}(x_t, x_{t+1})$ to be

$$\left\| \nabla_{x_{t+1}} \log P_F(x_{t+1}|x_t) - \nabla_{x_{t+1}} \log P_B(x_t|x_{t+1}) - \nabla_{x_{t+1}} \log F(x_{t+1}) \right\|^2, \qquad (7)$$

with the terminal flow loss on the logarithm scale

$$L_{\nabla\text{DB-terminal}}(x_T) = \left\| \nabla_{x_T} \log F(x_T) - \beta \nabla_{x_T} \log R(x_T) \right\|^2, \qquad (8)$$

where $\beta$ is a temperature coefficient and serve as a hyperparameter in the experiments. Notice that, by taking derivatives on the logarithms, we obtain the (conditional) score function $\nabla_{x_{t+1}} \log P_F(x_{t+1}|x_t)$ of the finetuned diffusion model. Indeed, the $\nabla$-DB loss is closely related to a Fisher divergence, also known as Fisher information score [25] (see Appendix B.1).

Similarly, by taking the derivative of both sides in Equation 5 with respect to $x_t$, one obtains the reverse $\nabla$-DB objective:

$$L_{\overleftarrow{\nabla}\text{DB}}(x_t, x_{t+1}) = \left\| \nabla_{x_t} \log P_F(x_{t+1}|x_t) - \nabla_{x_t} \log P_B(x_t|x_{t+1}) + \nabla_{x_t} \log F(x_t) \right\|^2. \quad (9)$$

Such $\nabla$-GFlowNet objectives constitute a valid GFlowNet algorithm (see the proof in Section B.2):

**Proposition 1.** *If $L_{\overrightarrow{\nabla}DB}(x_t, x_{t+1}) = L_{\overleftarrow{\nabla}DB}(x_t, x_{t+1}) = 0$ for any denoising transition $(x_t, x_{t+1})$ over the state space and $L_{\nabla DB\text{-terminal}}(x_T) = 0$ for all terminal state $x_T$, then the resulting forward policy generate samples $x_T$ with probability proportional to the reward function $R(x_T)^\beta$.*

**Remark 2.** The original detailed balance condition propagates information from the reward function to each state flow function in the sense of $F(x_{t+1}) \to (F(x_t), P_F(x_{t+1}|x_t))$, assuming the backward (noising) policy is fixed (*i.e.*, there is no learning component in the diffusion noising process). In our case, if we take a close look at Equation 7, we can see that $L_{\overrightarrow{\nabla}\text{DB}}(x_t, x_{t+1})$ could propagate the information from $F(x_{t+1})$ to the forward policy $P_F(x_{t+1}|x_t)$ but not $F(x_t)$.

**Remark 3.** Compared to previous GFlowNet works which use a scalar-output network to parameterize the (log-) flow function, in $\nabla$-GFlowNet we can directly use a U-Net [52]-like architecture that (whose output and input shares the same number of dimension) to parameterize $\nabla \log F(\cdot)$, which potentially provides more modeling flexibility. Furthermore, it is possible to initialize $\nabla \log F(\cdot)$ with layers from the pretrained model so that it can learn upon known semantic information.

## 3.2 RESIDUAL $\nabla$-DB FOR REWARD FINETUNING OF PRETRAINED MODELS

With the $\nabla$-DB losses, one can already finetune a diffusion model to sample from the reward distribution $R(x)$. However, the finetuned model may eventually over-optimize the reward and thus forget the pretrained prior (*e.g.*, how natural images look like). Instead, similar to other amortized inference work [82, 68], we consider the following objective with an augmented reward:

$$P_F(x_T) \propto R(x_T)^\beta P_F^\#(x_T), \qquad (10)$$

where $R(x)$ is the positive-valued reward function, $\beta$ is the temperature coefficient, $P_F^\#(x_T)$ is the marginal distribution of the pretrained model[4] and $P_F(x_T)$ is the marginal distribution of the finetuned model (as defined in Equation 1).

Because both the finetuned and pretrained model share the same backward policy $P_B$ (the noising process of the diffusion models), we can remove the $P_B$ term and obtain the forward *residual* $\nabla$-DB condition by subtracting the forward $\nabla$-DB equation for the pretrained model from the that of the finetuned model:

$$\underbrace{\nabla_{x_{t+1}} \log P_F(x_{t+1}|x_t) - \nabla_{x_{t+1}} \log P_F^\#(x_{t+1}|x_t)}_{\nabla_{x_{t+1}} \log \tilde{P}_F(x_{t+1}|x_t):\text{ residual policy score function}} = \underbrace{\nabla_{x_{t+1}} \log F(x_{t+1}) - \nabla_{x_{t+1}} \log F^\#(x_{t+1})}_{\nabla_{x_{t+1}} \log \tilde{F}(x_{t+1}):\text{ residual flow score function}}.$$

$$(11)$$

With the two residual terms defined above, we obtain the forward *residual* $\nabla$-DB objective:

$$L_{\overrightarrow{\nabla}\text{DB-res}}(x_t, x_{t+1}) = \left\| \nabla_{x_{t+1}} \log \tilde{P}_F(x_{t+1}|x_t) - \nabla_{x_{t+1}} \log \tilde{F}(x_{t+1}), \right\|^2. \qquad (12)$$

---

[4]We use the notation of $\#$ to indicate quantities of the pretrained model.

Similarly, we have the reverse *residual* $\nabla$-DB loss:

$$L_{\overleftarrow{\nabla}\text{DB-res}}(x_t, x_{t+1}) = \left\| \nabla_{x_t} \log \tilde{P}_F(x_{t+1}|x_t) + \nabla_{x_t} \log \tilde{F}(x_t) \right\|^2. \tag{13}$$

The terminal loss of the residual $\nabla$-DB method stays the same form as in Equation 8

$$L_{\nabla\text{DB-terminal}}(x_T) = \left\| \nabla_{x_T} \log \tilde{F}(x_T) - \beta \nabla_{x_T} \log R(x_T) \right\|^2. \tag{14}$$

**Proposition 4.** *If* $L_{\overrightarrow{\nabla}DB\text{-res}}(x_t, x_{t+1}) = L_{\overleftarrow{\nabla}DB\text{-res}}(x_t, x_{t+1}) = 0$ *for any denoising transition* $(x_t, x_{t+1})$ *over the state space and* $L_{\nabla DB\text{-terminal}}(x_T) = 0$ *for all terminal state* $x_T$, *then the resulting forward policy generate samples* $x_T$ *with probability proportional to* $R(x_T)^\beta P_F^{\#}(x_T)$.

**Remark 5.** We point out that perform the same way of deriving Equation 11, *i.e.*, subtraction between GFlowNet conditions from the finetuned and pretrained model, on the DB condition without gradient, we can obtain a *residual* DB condition $\tilde{F}(x_t)P_F(x_{t+1}|x_t) = P_F^{\#}(x_{t+1}|x_t)\tilde{F}(x_{t+1})$. Multiplying this condition across time and eliminate the term of intermediate $\tilde{F}(x_t)$ will lead to the objective derived in the relative GFlowNet work [67] as shown in Section B.4, which is a prior paper that proposes to work on reward finetuning GFlowNets with a given pretrained model.

**Remark 6.** One may completely eliminate the need for any residual flow score function with the *residual* $\nabla$-DB conditions of both directions: $\nabla_{x_{t+1}} \log \tilde{P}_F(x_{t+1}|x_t) = -\nabla_{x_{t+1}} \log \tilde{P}_F(x_{t+2}|x_{t+1})$. The bidirectional *residual* $\nabla$-DB condition can be analogously understood as the balance condition of two forces from $x_t$ and $x_{t+2}$ acting on $x_{t+1}$: if not balanced, one can locally find some other $x_{t+1}$ that makes both transitions more probable.

**Flow reparameterization through forward-looking (FL) trick.** Though mathematically valid, the bidirectional pair of $\nabla$-DB conditions suffers from inefficient credit assignment for long sequences, a problem commonly observed in RL settings [63, 66]. Instead, we may leverage the priors we have from the pretrained diffusion model to speed up the finetuning process and consider the individual conditions for the forward and reverse directions. Specifically, we employ the forward-looking (FL) technique for GFlowNets [44, 80] and parameterize the residual flow score function with a "baseline" of the "one-step predicted reward gradient":

$$\nabla_{x_t} \log \tilde{F}(x_t) \triangleq \beta \gamma_t \nabla_{x_t} \log \underbrace{R(\hat{x}_\theta(x_t))}_{\text{predicted reward}} + g_\phi(x_t) \tag{15}$$

where $\gamma_t$ is the scalar to control the strength of forward looking with the constraint $\gamma_T = 1$ and $g_\phi(x_t)$ is the *actual* neural network with parameters satisfying the terminal constraint $g_\phi(x_T) = 0$. Here $\hat{x}_\theta(\cdot)$ is the one-step clean data prediction defined in Equation 2. With the FL technique, one achieves faster convergences since it sets a better initialization for the residual flow score function than a naïve zero or random initialization [44].

We therefore obtain the forward-looking version of *residual* $\nabla$-DB losses of both directions:

$$L_{\overrightarrow{\nabla}\text{DB-FL-res}}(x_t, x_{t+1}) = \left\| \nabla_{x_{t+1}} \log \tilde{P}_F(x_{t+1}|x_t; \theta) - \left[ \beta \gamma_t \nabla_{x_{t+1}} \log R(\hat{x}_\theta(x_{t+1})) + g_\phi(x_{t+1}) \right] \right\|^2. \tag{16}$$

$$L_{\overleftarrow{\nabla}\text{DB-FL-res}}(x_t, x_{t+1}) = \left\| \nabla_{x_t} \log \tilde{P}_F(x_{t+1}|x_t) + \left[ \beta \gamma_t \nabla_{x_t} \log R(\hat{x}_\theta(x_t)) + g_\phi(x_t) \right] \right\|^2. \tag{17}$$

Moreover, the corresponding terminal loss objective now becomes

$$L_{\nabla\text{DB-FL-terminal}}(x_T) = \left\| \nabla_{x_T} \log \tilde{F}(x_T) - \beta \gamma_t \nabla_{x_T} \log R(x_T) \right\|^2 = \left\| g_\phi(x_T) \right\|^2, \tag{18}$$

which indicates that the actual parameterized flow network $g_\phi$ should take a near-zero value for terminal states $x_T$. The total loss on a collected trajectory $\tau = (x_0, x_1, ..., x_T)$ is therefore

$$\sum_t \left[ w_F(t) L_{\overrightarrow{\nabla}\text{DB-FL-res}}(x_t, x_{t+1}) + w_B(t) L_{\overleftarrow{\nabla}\text{DB-FL-res}}(x_t, x_{t+1}) \right] + L_{\nabla\text{DB-FL-terminal}}(x_T) \tag{19}$$

where $w_F(t)$ and $w_B(t)$ are scalar weights to control the relative importance of each term.

We summarize the resulting algorithm in Algorithm 1 in Appendix 1.

**Choice of FL scale.** Naïvely setting $\gamma_t = 1$ can be aggressive especially when the reward scale $\beta$ and the learning rate are set to a relatively high value. Inspired by the fact that in diffusion models $F_t(x_t)$ can be seen as $R(x)$ smoothed with a Gaussian kernel, we propose to set $\gamma_t = \alpha_{T-t}$.

## 4 Experiments and Results

### 4.1 Baselines

For gradient-free methods, we consider DAG-DB [80] (*i.e.*, GFlowNet finetuning with the DB objective) and DDPO [3]. Since the original DB objective aims to finetune with $R^\beta(x)$ instead of $P_F^{\#}(x_T)R^\beta(x_T)$, we also consider the forward-looking residual DB loss, defined as

$$L_{\text{DB-FL-res}}(x_t, x_{t+1}) = \Big( \log \tilde{P}_F(x_{t+1}|x_t) - \beta \log R(\hat{x}_\theta(x_{t+1})) - g_\phi(x_{t+1})$$
$$+ \beta \log R(\hat{x}_\theta(x_t)) + g_\phi(x_t) \Big)^2. \tag{20}$$

For other gradient-aware finetuning methods, we consider ReFL [71] and DRaFT [6]. ReFL samples a trajectory and stops at some random time step $t$, with which it maximizes $R(\hat{x}_\theta(x_t)$ where $\hat{x}_\theta(\cdot)$ is the one-step sample prediction function and $x_t$ is the computed via denoising $\overleftarrow{\nabla}(x_{t-1})$ with $\overleftarrow{\nabla}$ being the stop-gradient operation. Differently, DRaFT samples time step $T - K$ (typically $K = 1$) and expand the computational graph of DDPM from $\overleftarrow{\nabla}(x_{T-K})$ to $x_T$ so that $R(x_T)$ can be backpropagated to $x_{T-K}$, with all $x_t$ in the previous time steps removed from this computational graph. A variant of DRaFT called DRaFT-LV performs a few more differentiable steps of "noising-denoising" on the sampled $x_T$ before feeding it into the reward function $R(\cdot)$. We follow the paper of DRaFT and use only one "noising-denoising" step: $x'_{T-1} \sim P_B(x'_{T-1}|x_T)$ and $x'_T \sim P_F(x'_T|x'_{T-1})$.

### 4.2 Reward functions, prompt datasets and metrics

For the main experiments, we consider two reward functions: Aesthetic Score [28], Human Preference Score (HPSv2) [69, 70] and ImageReward [71], all of which trained on large-scale human preference datasets such as LAION-aesthetic [28] and predict the logarithm of reward values. For base experiments with Aesthetic Score, we use a set of 45 simple animal prompts as used in DDPO [3]; for those with HPSv2, we use photo+painting prompts from the human preference dataset (HPDv2) [69]. To measure the diversity of generated images, we follow Domingo-Enrich et al. [10] and compute the variance of latent features extracted from a batch of generated images (we use a batch of size 64). Using the same set of examples, we evaluate the capability of prior preservation, we compute the per-prompt FID score between images generated from the pretrained model and from the finetuned model and take the average FID score over all evaluation prompts.

### 4.3 Experiment settings

For all methods, we use 50-step DDPM sampler [17] to construct the MDP. StableDiffusion-v1.5 [51] is used as the base model. For finetuning diffusion model policies, we use LoRA [21] with rank 8. The residual flow score function in *residual* $\nabla$-DB is set to be a scaled-down version of the StableDiffusion U-Net, whereas the flow function (in DAG-DB and *residual* DB) is set to be a similar network but without the U-Net decoding structure (since the desired output is a scalar instead of an image vector). Both networks are initialized with tiny weights in the final output layers.

As the landscape of $R(x)$ can be highly non-smooth, we approximate $\nabla_{x_t} \log R(\hat{x}_\theta(x_t))$ with $\mathbb{E}_{\epsilon \sim \mathcal{N}(0,c)} \nabla_{x_t} \log R(\hat{x}_\theta(x_t) + \epsilon)$ where $c$ is a tiny constant. For StableDiffusion [51], since the diffusion process runs in the latent space, the reward function is instead $\mathbb{E}_{\epsilon \sim \mathcal{N}(0,c)} \nabla_{x_t} \log R(\text{decode}(\hat{x}_\theta(x_t) + \epsilon))$ in which decode$(\cdot)$ is the pretrained (and frozen) VAE decoder and $c$ is set to $2 \times 10^{-3}$, slightly smaller than one pixel (i.e., $1/255$). We approximate this expectation with 3 independent samples for each transition in each trajectory. For all experiments, we try 3 random seeds. We set $w_F(t) = 1$ for all $t$'s and unless otherwise specified $w_B(t) = 1$.

To stabilize the training process of our method (*residual* $\nabla$-DB), we follow the official repo of DAG-DB [80] and uses the following output regularization: $\lambda \|\epsilon_\theta(x_t) - \epsilon_{\theta^\dagger}(x_t)\|^2$ where $\theta^\dagger$ is the diffusion model parameters in the previous update step[5]. For *residual* $\nabla$-DB, We set the output regularization strength $\lambda = 2000$ in Aesthetic Score experiments and $\lambda = 5000$ in HPSv2 and ImageReward experiments. For all experiments with *residual* $\nabla$-DB, we set the learning rate to

---

[5]Essentially a Fisher divergence between the pretrained and the finetuned distributions conditioned on $x_t$. Similar regularizations with KL divergence has been seen in RL algorithms like TRPO [54] and PPO [55].

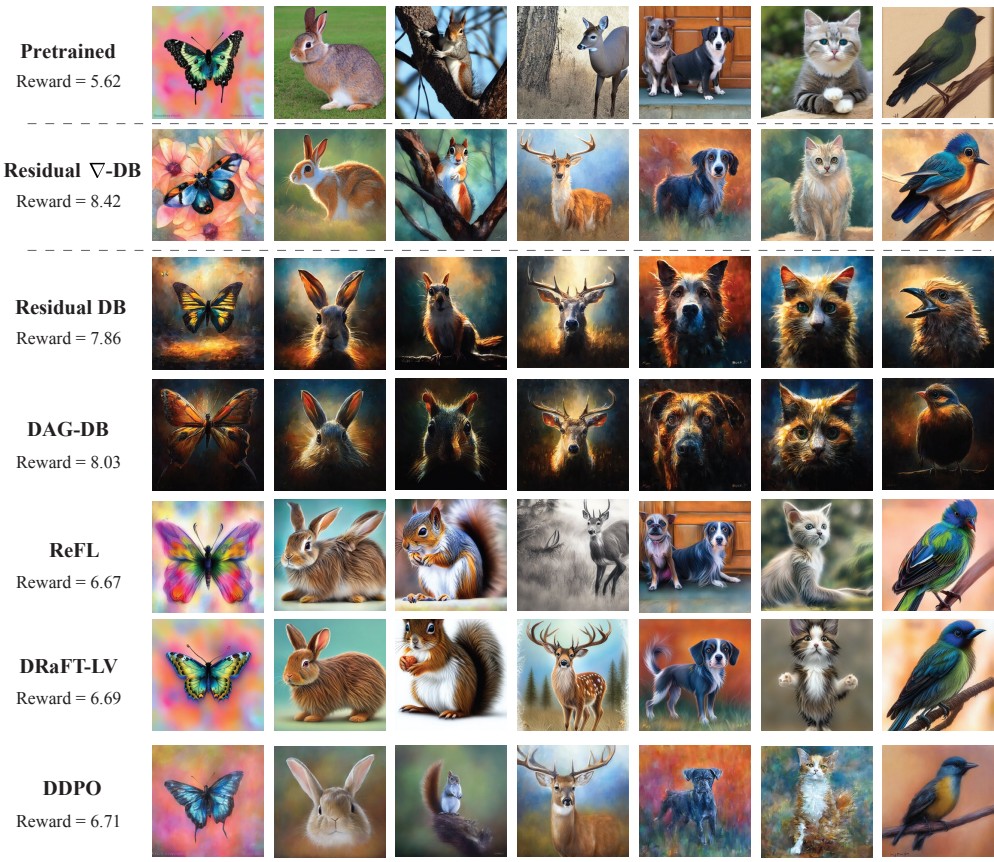

Figure 2: Comparison between images generated by models finetuned with different methods for a maximum of 200 update steps. For each method, we pick the model trained that produces images with the highest rewards without semantic collapse among all model checkpoints, as methods like ReFL and DRaFT-LV easily collapses (as illustrated in Fig. 3). For each method, we show the average reward of the corresponding presented images.

$1 \times 10^{-3}$ and ablate over a set of choices of reward temperature $\beta$, in a range such that the reward gradients are more significant than the residual policy score function $\nabla_{x_t} \log P_F(x_{t+1}|x_t)$ of the pretrained model. For HPSv2 and ImageReward experiments, we set $\beta$ to be 500000 and 10000, respectively. For each epoch, we collect 64 generation trajectories for each of which we randomly shuffle the orders of transitions. We use the number of gradient accumulation steps to 4 and for each 32 trajectories we update both the forward policy and the residual flow score function. For *residual* $\nabla$-DB in most of the experiments, we for training sub-sample 10% of the transitions in each collected trajectory by uniformly sample one transition in each of the uniformly split time-step intervals but ensuring that the final transition step always included.

For *residual* DB and DAG-DB, we set the learning rate to $3 \times 10^{-4}$ with the output regularization strength $\lambda = 1$ The sampling and training procedures are similar to the *residual* $\nabla$-DB experiments. For both ReFL and DRaFT, we use a learning rate of $10^{-4}$. For ReFL, we follow the official repo and similarly set the random stop time steps to between 35 and 49. For DRaFT, since the official code is not released, we follow the settings in AlignProp [48], a similar concurrent paper. We set the loss for both ReFL and DRaFT to $-\mathbb{E}_{x_T \sim P_F} \text{ReLU}(R(\text{decode}(x_T)))$ where the ReLU function is introduced for training stability in the case of the ImageReward reward function.

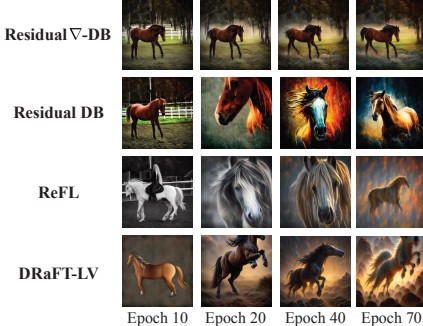

Figure 3: Our $\nabla$-GFlowNet finetuning yields stable output compared to other baselines.

## 4.4 EXPERIMENTAL RESULTS

**General experiments.** In Figure 6 and Table 1, we show the evolution of reward, DreamSim diversity and FID scores of all methods with the mean curves and the corresponding standard deviations

| Pretrained | Residual $\nabla$-DB | Residual DB | DDPO | ReFL | DRaFT-LV |

Prompt: The image features a castle surrounded by a dreamy garden with roses and a cloudy sky in the background.

Prompt: A painting of a Bladerunner interior room in Africa with detailed artwork.

Prompt: A broken videogame console with a colorful and compelling painting.

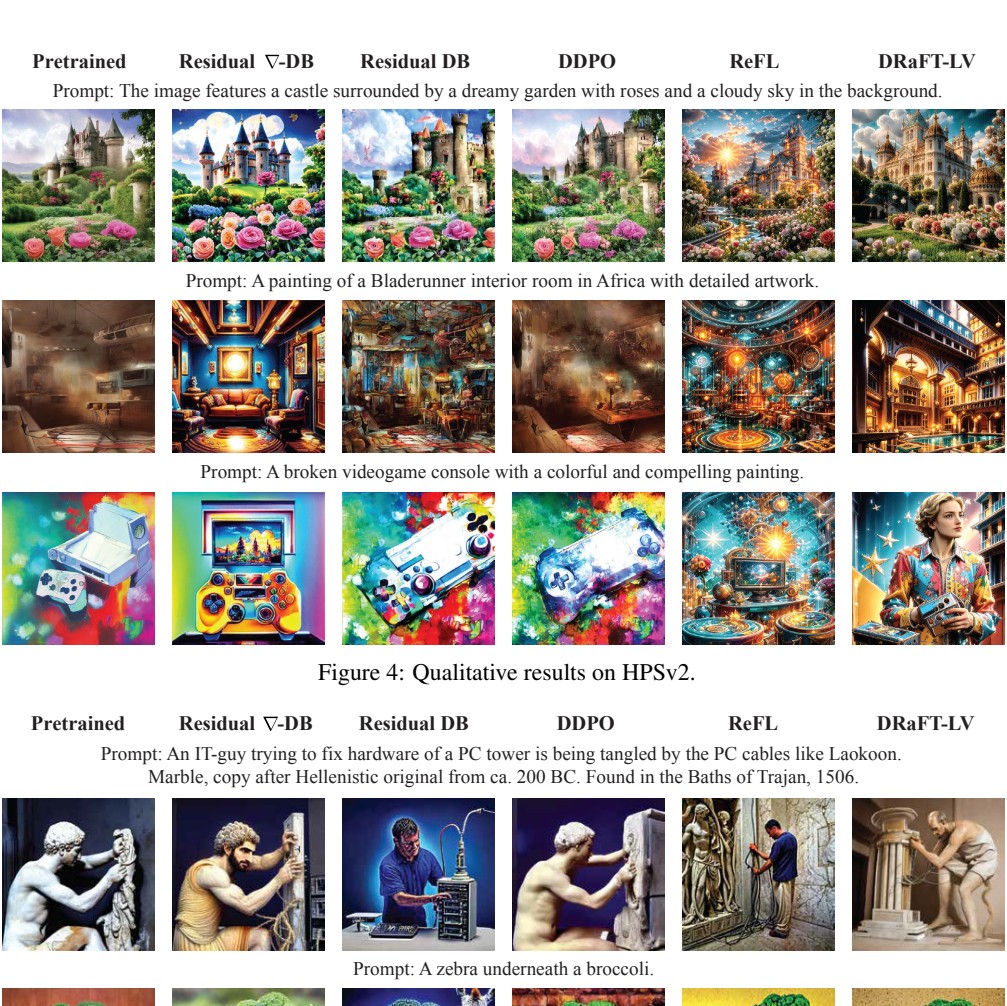

Figure 4: Qualitative results on HPSv2.

| Pretrained | Residual $\nabla$-DB | Residual DB | DDPO | ReFL | DRaFT-LV |

Prompt: An IT-guy trying to fix hardware of a PC tower is being tangled by the PC cables like Laokoon.
Marble, copy after Hellenistic original from ca. 200 BC. Found in the Baths of Trajan, 1506.

Prompt: A zebra underneath a broccoli.

Prompt: A blue colored dog.

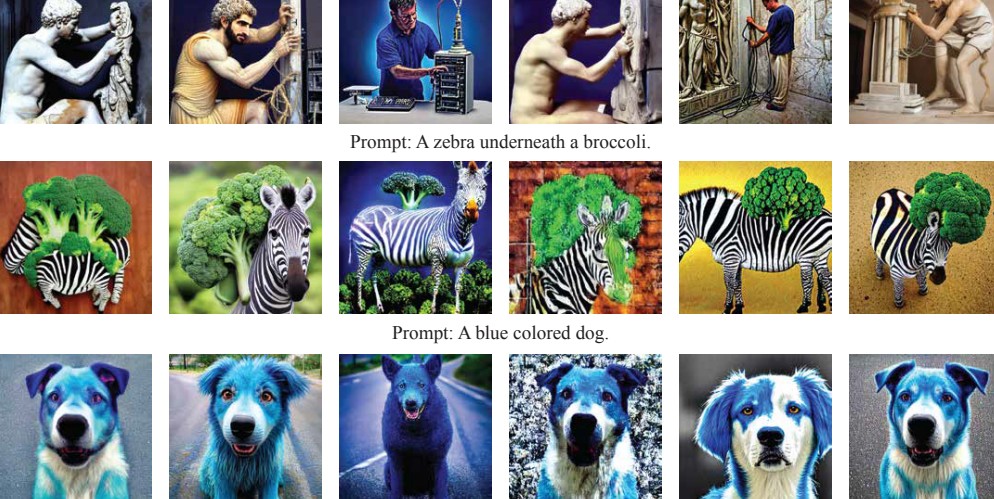

Figure 5: Qualitative results on ImageReward.

(on 3 random seeds). Our proposed *residual* $\nabla$-DB is able to achieve comparable convergence speed, measured in update steps, to that of the gradient-free baselines while those diversity-aware baselines fail to do so. In Figure 7, we plot the diversity-reward tuples and FID-reward tuples for models evaluated at different checkpoints (every 5 update steps) and show that our method achieves Pareto improvements on diversity preservation, prior preservation and reward. The gradient-informed baselines, ReFL and the DRaFT variants, generally behave worse than *residual* $\nabla$-DB due to their mode-seeking nature. Qualitatively, we show that the model finetuned with *residual* $\nabla$-DB on Aesthetic Score generate more aesthetic and more diverse samples in both style and subject identity (Fig. 2 and Fig. 24), while the other baselines exhibit mode collapse or even catastrophic forgetting of pre-trained image prior. We also demonstrate some images generated by the diffusion model finetuned with *residual* $\nabla$-DB on HPSv2 in Figure 4 and on ImageReward in Figure 5. Furthermore, we qualitatively show that *residual* $\nabla$-DB is robust while with the gradient-informed baseline methods are prone to training collapse (Fig. 3). Due to limited space, here we only show the most important abla-

| Method | Aesthetic Score | | | HPSv2 | | | ImageReward | | |
|---|---|---|---|---|---|---|---|---|---|
| | Reward ($\uparrow$) | Diversity DreamSim ($\uparrow, 10^{-2}$) | FID ($\downarrow$) | Reward ($\uparrow, 10^{-1}$) | Diversity DreamSim ($\uparrow, 10^{-2}$) | FID ($\downarrow$) | Reward ($\uparrow, 10^{-1}$) | Diversity DreamSim ($\uparrow, 10^{-2}$) | FID ($\downarrow$) |
| Base Model | $5.83 \pm 0.01$ | $35.91 \pm 0.00$ | $216 \pm 1$ | $2.38 \pm 0.13$ | $37.75 \pm 0.21$ | $563 \pm 5$ | $-0.38 \pm 0.12$ | $41.09 \pm 0.03$ | $468 \pm 1$ |
| DDPO | $6.68 \pm 0.14$ | $32.96 \pm 1.04$ | $\mathbf{312} \pm 9$ | $2.52 \pm 0.04$ | $3.49 \pm 0.03$ | $\mathbf{681} \pm 16$ | $0.27 \pm 0.38$ | $38.51 \pm 1.49$ | $714 \pm 25$ |
| ReFL | $9.53 \pm 0.46$ | $8.20 \pm 3.06$ | $1765 \pm 51$ | $3.67 \pm 0.06$ | $19.84 \pm 1.70$ | $1191 \pm 46$ | $1.36 \pm 0.30$ | $36.50 \pm 0.52$ | $597 \pm 10$ |
| DRaFT-1 | $10.16 \pm 0.13$ | $4.24 \pm 0.45$ | $1665 \pm 182$ | $3.70 \pm 0.06$ | $18.96 \pm 1.35$ | $1222 \pm 84$ | $1.59 \pm 0.25$ | $37.27 \pm 0.49$ | $531 \pm 13$ |
| DRaFT-LV | $\mathbf{10.21} \pm 0.34$ | $6.39 \pm 1.66$ | $1854 \pm 296$ | $\mathbf{3.75} \pm 0.08$ | $21.13 \pm 1.19$ | $1164 \pm 43$ | $1.44 \pm 0.25$ | $37.56 \pm 0.09$ | $529 \pm 18$ |
| DAG-DB | $7.73 \pm 0.07$ | $15.88 \pm 0.70$ | $595 \pm 87$ | $2.52 \pm 0.06$ | $\mathbf{32.50} \pm 0.59$ | $866 \pm 41$ | $4.70 \pm 0.75$ | $26.78 \pm 0.64$ | $809 \pm 34$ |
| *residual* DB | $7.20 \pm 0.92$ | $19.38 \pm 4.07$ | $1065 \pm 587$ | $2.55 \pm 0.06$ | $32.49 \pm 0.47$ | $840 \pm 107$ | $\mathbf{6.47} \pm 0.54$ | $25.52 \pm 0.35$ | $772 \pm 19$ |
| $\nabla$-GFlowNet ($w_B=0$) | $7.86 \pm 0.06$ | $29.21 \pm 0.18$ | $318 \pm 13$ | $3.53 \pm 0.03$ | $24.40 \pm 0.56$ | $973 \pm 5$ | $5.33 \pm 0.62$ | $35.85 \pm 0.66$ | $638 \pm 15$ |
| $\nabla$-GFlowNet ($w_B=1$) | $7.90 \pm 0.09$ | $\mathbf{29.67} \pm 0.51$ | $317 \pm 15$ | $3.53 \pm 0.04$ | $24.39 \pm 0.87$ | $1000 \pm 39$ | $2.23 \pm 0.34$ | $\mathbf{39.85} \pm 0.67$ | $\mathbf{501} \pm 7$ |

Table 1: Comparison between the models finetuned with our proposed method $\nabla$-GFlowNet (with the objective *residual* $\nabla$-DB) and the baselines. All models are finetuned with 200 update steps. For each method, the model with the best mean reward is used for evaluation. Note that while baselines like DDPO can achieve better scores on some metric, it often comes with the price of much worse performance on some other.

tion studies and leave the rest to Appendix J. For the same reason, we leave plots for experiments on HPSv2 and ImageReward to Appendix I and qualitative comparisons on diversity to Appendix M.

**Effect of reward temperature.** We perform ablation study on Aesthetic Score with $\beta \in \{5000, 7000, 10000\}$ in *residual* $\nabla$-DB. Not surprisingly, a higher reward temperature leads to faster convergence at the cost of worse diversity- and worse prior-preservation, as observed in Figure 8.

**Effect of sub-sampling.** Typically, sub-sampling results in worse gradient estimates. We empirically study how sub-sampling may effect the performance and show the results in Figure 10 and 11 in the appendix. We empirically do not observe huge performance drop due to the subsampling strategy, potentially because the rich gradient signals in both the reward and the flow are sufficient.

**Effect of attenuating scaling on predicted reward.** In Fig. 16 and 17, we show that the model trained without attenuation converges faster, but at the cost of worse diversity and prior-preservation.

**Reward finetuning with other sampling algorithms.** To show that our method generalizes to different sampling diffusion algorithms, we construct another MDP based on SDE-DPM-Solver++ [38], with 20 inference steps. In Fig. 18 and 19, we obverse that our *residual* $\nabla$-DB can still achieve a good balance between reward convergence speed, diversity preservation and prior preservation.

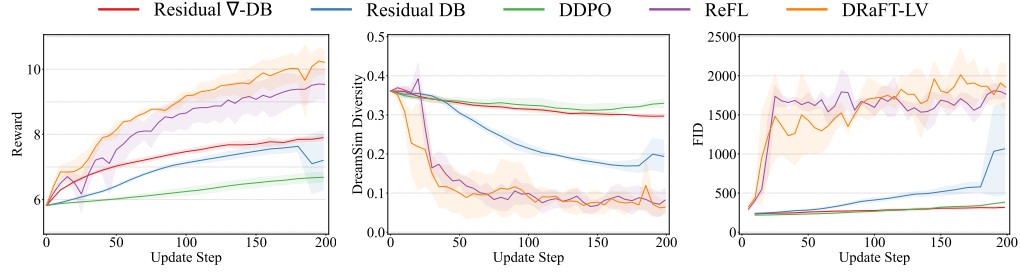

Figure 6: Convergence curves of different metrics for different methods throughout the finetuning process on Aesthetic Score. Finetuning with our proposed *residual* $\nabla$-DB converges faster than the non-gradient-informed methods and with better diversity- and prior-preserving capability.

## 5 RELATED WORK

**Reward finetuning of diffusion models.** The demand for reward finetuning is commonly seen in alignment, where one obtains utilize a human reference reward function to align the behavior of generative models [22, 1, 69] for better instruction following capability and better AI safety. With a reward function, typically obtained by learning from human preference datasets [84, 62], one may use reinforcement learning (RL) algorithms, for instance PPO [55], to adapt not only autoregressive language models [43] which naturally admits a Markov decision process (MDP), but also diffusion

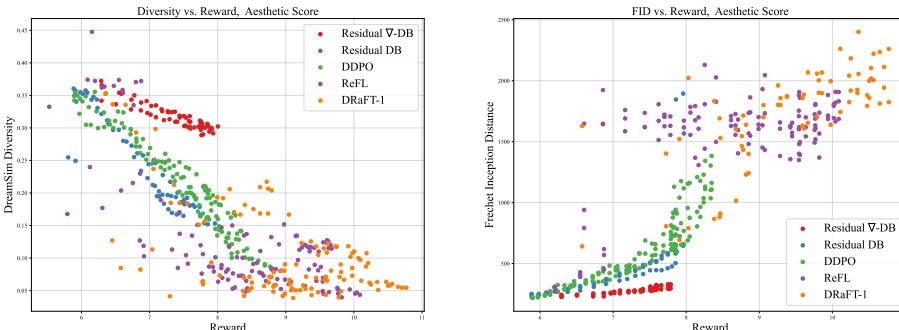

Figure 7: Trade-offs between reward, diversity preservation and prior preservation for different reward fine-tuning methods. Dots represent the evaluation results of models checkpoint saved after every 5 iterations of finetuning, where ones with larger reward, larger diversity scores and smaller FID scores are considered better.

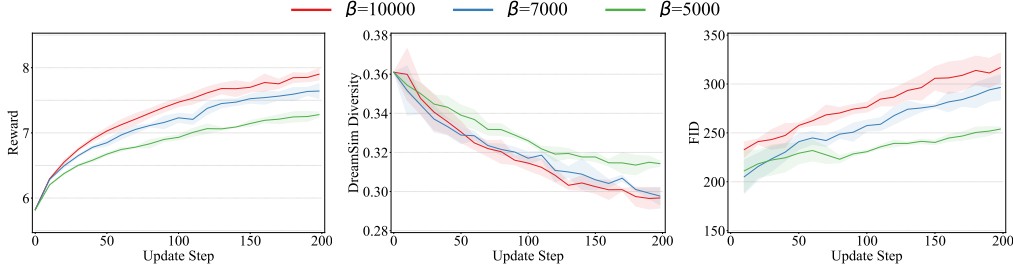

Figure 8: Higher temperature $\beta$ leads to faster convergence but with less diversity and less prior preservation.

models [3, 12]. Specifically, one can construct MDPs from some diffusion sampling algorithm [59, 38] by considering each noisy image at some inference step as a state and each denoising step as an action. Besides RL algorithms, there exist some other approaches, including stochastic optimal control [10, 65], GFlowNets [80] and some other ones akin to RL methods [30, 11]. While most of the aforementioned approaches train with only black-box rewards, once we have access to a differentiable reward function we may accelerate the finetuning process with reward gradient signals. For instance, methods exist to construct a computational graph from sampled generation trajectories to directly optimize for rewards [6, 48, 71], yet with these methods models are not trained to correctly sample according to the reward function. While one may also generate samples from the reward function without finetuning using plug-in guidance methods for diffusion models [9, 60, 26, 14] as an alternative, but the generated distributions are often very biased. Besides, reward finetuning for diffusion models is typically memory consuming as many methods require a large computational graph rolled out from long generation trajectories, for which it is typical to employ parameter-efficient finetuning techniques [21, 49, 33].

**GFlowNets.** Generative flow network (GFlowNet) [2] is a high-level algorithmic framework that introduces sequential decision-making into generative modeling [75], bridging methodology between reinforcement learning [79, 45, 46, 44, 29] and energy-based modeling [76]. GFlowNets are versatile since many models, such as diffusion models, can be easily treated as specifications of GFlowNets [75, 27, 78, 56]. GFlowNets perform amortized variational inference [40] and generate samples with probability proportional to a given density or reward. GFlowNets are used in various applications including but not limited to drug discovery [23, 24, 57], structure learning [8], phylogenetic inference [83], combinatorial optimization [74, 77], and prompt adaptation [73].

## 6 CONCLUDING REMARKS

We propose $\nabla$-GFlowNet, a fast, diversity and prior-preserving diffusion alignment method, by leveraging gradient information in the probabilistic framework of GFlowNets that aims to sample according to a given unnormalized density function or reward. Specifically, we develop $\nabla$-DB, the gradient-informed version of the Detailed Balance objective, and the variant of *residual* $\nabla$-DB with which one may finetune a diffusion model with reward in a prior-preserving way. Our empirical results show that $\nabla$-GFlowNet achieves a better trade-off between convergence speed, diversity in generated samples and prior preservation. We hope that our method sheds lights on future studies on more efficient post-training alignment strategies of diffusion models as well as related applications.

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

# Appendix

## Table of Contents

# A  OVERALL ALGORITHM

---

**Algorithm 1** $\nabla$-GFlowNet Diffusion Finetuning with *residual* $\nabla$-DB

---

1: **Inputs:** Pretrained diffusion model $f_{\theta\#}$ with the MDP constructed from some Markovian sampler, reward function $R(\cdot)$

2: **Initialization:** Model to finetune $f_\theta$ with $\theta = \theta^\#$, residual flow score function $g_\phi(\cdot)$.

3: Sample the initial batch of trajectories $\mathcal{D}_{\text{prev}} = \{(x_1, ..., x_T)_i\}_{i=1...N}$ with the current finetuned diffusion model $f_\theta$.

4: Set $\theta^\dagger \leftarrow \theta$.

5: **while** not converged **do**

6:    Sample a batch of trajectories $\mathcal{D}_{\text{curr}} = \{(x_1, ..., x_T)_i\}_{i=1...N}$ with the finetuned diffusion model.

7:    Subsample the time steps to train with: the full set $\mathcal{T}_i = \{1, ..., T\}$ or the sampled set $\mathcal{T}_i = \text{Sample-N}(\{1, ..., T\})$ where Sample-N is some unbiased sampling algorithm to randomly pick $N$ samples.

8:    Compute the loss
$\sum_{t \in \mathcal{T}_i, (x_{1:T})_i \in \mathcal{D}_{\text{prev}}} L_{\nabla\text{DB-FL-res}}(x_t, x_{t+1}; \theta, \theta^\#, \phi, R)$
$+ L_{\nabla\text{DB-FL-terminal}}(x_T; \phi) + \lambda\|f_\theta(x_t) - f_{\theta^\dagger}(x_t)\|^2$.

9:    Set $\theta^\dagger \leftarrow \theta$.

10:   Update the diffusion model and the residual flow score function.

11:   Set $\mathcal{D}_{\text{prev}} \leftarrow \mathcal{D}_{\text{curr}}$.

12: **end while**

13: **return** finetuned model $f_\theta$.

---

# B  ALGORITHMIC DETAILS

## B.1  $\nabla$-DB OBJECTIVE AS A STATISTICAL DIVERGENCE

$L_{\nabla\text{DB}}$ (Equation 7) is analogous to a Fisher divergence (up to a constant scale) if we always use on-policy samples to update the diffusion model and the flow function:

$$
D_{\text{Fisher}}\Big(P_F(x_{t+1}|x_t)\Big|\Big|\frac{P_B(x_t|x_{t+1})F(x_{t+1})}{F(x_t)}\Big)
$$

$$
= \frac{1}{2}\mathop{\mathbb{E}}_{x_{t+1}\sim P_F(x_{t+1}|x_t)}\Big|\Big|\nabla_{x_{t+1}}\log P_B(x_t|x_{t+1}) - \nabla_{x_{t+1}}\log\frac{P_B(x_t|x_{t+1})F(x_{t+1})}{F(x_t)}\Big|\Big|^2
$$

$$
= \frac{1}{2}\mathop{\mathbb{E}}_{x_{t+1}\sim P_F(x_{t+1}|x_t)} L_{\nabla\text{DB}}(x_t, x_{t+1}). \tag{21}
$$

## B.2  PROOF OF PROPOSITION 1

*Proof.* When the training objectives equal $0$ for all states, we would have

$$
\nabla_{x_{t+1}}\log P_F(x_{t+1}|x_t) = \nabla_{x_{t+1}}\log P_B(x_t|x_{t+1}) + \nabla_{x_{t+1}}\log F_{t+1}(x_{t+1}) \tag{22}
$$

$$
\nabla_{x_t}\log P_F(x_{t+1}|x_t) = \nabla_{x_t}\log P_B(x_t|x_{t+1}) - \nabla_{x_t}\log F_t(x_t) \tag{23}
$$

$$
\nabla_{x_T}\log F(x_T) = \beta\nabla_{x_T}\log R(x_T) \tag{24}
$$

for any trajectory $(x_0,\ldots,x_T)$.

Through indefinite integral, these indicate that there exist a function $C_t(x_t)$ satisfies

$$
C_t(x_t)P_F(x_{t+1}|x_t) = F_{t+1}(x_{t+1})P_B(x_t|x_{t+1}) \tag{25}
$$

$$
F_t(x_t)P_F(x_{t+1}|x_t) = C_{t+1}(x_{t+1})P_B(x_t|x_{t+1}) \tag{26}
$$

$$
F(x_T) \propto R(x_T)^\beta. \tag{27}
$$

Therefore, we have

$$
\frac{C_t(x_t)}{F_t(x_t)} = \frac{F_{t+1}(x_{t+1})}{C_{t+1}(x_{t+1})}, \quad \forall(x_t, x_{t+1}). \tag{28}
$$

The right hand side does not depend on $x_t$, therefore, the left hand side is a constant. So we have

$$
C_t(x_t) \propto F_t(x_t), \quad \forall t. \tag{29}
$$

The probability of generating a data $x_T$ then equals

$$
P_F(x_T) = \int P_0(x_0|\oslash)\prod_t P_F(x_{t+1}|x_t)dx_{0:T-1} \tag{30}
$$

$$
= \int F_0(x_0)\prod_t \frac{F_{t+1}(x_{t+1})P_B(x_t|x_{t+1})}{C_t(x_t)}dx_{0:T-1} \tag{31}
$$

$$
\propto \int F_0(x_0)\prod_t \frac{F_{t+1}(x_{t+1})P_B(x_t|x_{t+1})}{F_t(x_t)}dx_{0:T-1} \tag{32}
$$

$$
\propto F(x_T)\int\prod_t P_B(x_t|x_{t+1})dx_{0:T-1} \tag{33}
$$

$$
\propto F(x_T) \propto R(x_T)^\beta, \tag{34}
$$

which proves the validity of the $\nabla$-GFlowNet algorithm. $\qquad\square$

## B.3  PROOF OF PROPOSITION 4

When the training objectives equal $0$ for all states, we have

$$
\nabla_{x_{t+1}}\log\tilde{P}_F(x_{t+1}|x_t) = \nabla_{x_{t+1}}\log\tilde{F}(x_{t+1}) \tag{35}
$$

$$
\nabla_{x_t}\log\tilde{P}_F(x_{t+1}|x_t) = -\nabla_{x_t}\log\tilde{F}(x_t) \tag{36}
$$

$$
\nabla_{x_T}\log\tilde{F}(x_T) = \beta\nabla_{x_T}\log R(x_T) \tag{37}
$$

for any trajectory $(x_0, \dots, x_T)$.

Through indefinite integral, these indicate that there exist a function $C_t(x_t)$ satisfies

$$C_t(x_t)\tilde{P}_F(x_{t+1}|x_t) = \tilde{F}(x_{t+1}) \tag{38}$$

$$\tilde{F}(x_t)\tilde{P}_F(x_{t+1}|x_t) = C_{t+1}(x_{t+1}) \tag{39}$$

$$\tilde{F}(x_T) \propto R(x_T)^\beta. \tag{40}$$

Thus we have

$$\frac{C_t(x_t)}{\tilde{F}_t(x_t)} = \frac{\tilde{F}_{t+1}(x_{t+1})}{C_{t+1}(x_{t+1})}, \quad \forall(x_t, x_{t+1}). \tag{41}$$

The right hand side does not depend on $x_t$, therefore, the left hand side is a constant. So we have

$$C_t(x_t) \propto \tilde{F}_t(x_t), \quad \forall t. \tag{42}$$

The probability of generating a data $x_T$ then equals

$$P_F(x_T) = \int P_0(x_0|\oslash) \prod_t P_F(x_{t+1}|x_t) dx_{0:T-1} \tag{43}$$

$$= \int P_0^\#(x_0|\oslash) \prod_t P_F^\#(x_{t+1}|x_t) \prod_t \tilde{P}_F(x_{t+1}|x_t) dx_{0:T-1} \tag{44}$$

$$= \int P_0^\#(x_0|\oslash) \prod_t P_F^\#(x_{t+1}|x_t) \frac{\tilde{F}_{t+1}(x_{t+1})}{C_t(x_t)} dx_{0:T-1} \tag{45}$$

$$\propto \tilde{F}_T(x_T) \int P_0^\#(x_0|\oslash) \prod_t P_F^\#(x_{t+1}|x_t) dx_{0:T-1} \tag{46}$$

$$\propto R(x_T)^\beta P_F^\#(x_T). \tag{47}$$

Hence zero $\nabla$-DB losses in both forward and reverse direction for all $(x_t, x_{t+1})$ pairs yield the ideal tilted distribution $R(x_T)^\beta P_F^\#(x_T)$.

### B.4 Relationship between residual DB and Trajectory Balance

Here we illustrate the Remark 5. A different but equivalent condition for GFlowNets states:

**Trajectory Balance (TB) [39].** The following TB condition must hold for any transition sequence $(s_0, s_1, ..., s_N)$ where $x_0$ is the unique starting state in the MDP and $s_N$ is a terminal state, given a GFlowNet with the forward policy $P_F(s'|s)$ and the backward policy $P_B(s|s')$:

$$\log \frac{Z \prod P_F(s'|s)}{R(x_T) \prod P_B(s|s')} = 0 \tag{48}$$

where $Z = F(x_0)$ is the total flow and $R(x_T) = F(x_T)$ the reward. The proof is immediate with a telescoping product of the DB condition.

With an (ideal) pretrained model $P_F^\#$ and the satisfication of the finetuning objective of Equation 10, one can prove the conclusion in [67]:

$$\log \frac{Z \prod P_F(s'|s)}{Z^\# \prod P_F^\#(s'|s)} = \beta \log R(x_T), \tag{49}$$

which is also an immediate result of a telescoping products of the *residual* DB condition (which leads to Equation 20):

$$\log \tilde{P}_F(x_{t+1}|x_t) = \log \tilde{F}(x_{t+1}) - \log \tilde{F}(x_t). \tag{50}$$

While Equation 49 and residual DB are mathematically equivalent, implementation-wise TB in Equation 49 demands the whole sampling sequence be stored in the memory for gradient computation, or one has resort to the time-costly technique of gradient checkpointing. In comparison, with DB-based methods one may amortize the computational cost into flows at different time steps and therefore allow diffusion finetuning with flexible sampling sequence, of which the distribution approximation capacity and generation performance are generally greater.

## C  GENERALIZATION TO THE LEARNABLE $P_B$ SETTING

If the backward process $P_B$ is learnable, the resulted $P_B$ does not cancel the backward propcess $P_B^\#$ in the pretrained model in the derivation of the *residual* $\nabla$-DB condition. Let $\log \tilde{P}_B(x_t|x_{t+1};\theta_B) = \log P_B(x_t|x_{t+1};\theta_B) - \log P_B^\#(x_t|x_{t+1})$. We instead have the following *residual* $\nabla$-DB losses:

$$\nabla_{x_{t+1}} \log \tilde{P}_F(x_{t+1}|x_t;\theta) - \nabla_{x_{t+1}} \log \tilde{P}_B(x_t|x_{t+1};\theta_B) = \nabla_{x_{t+1}} \log \tilde{F}(x_{t+1};\phi) \tag{51}$$

$$\nabla_{x_t} \log \tilde{P}_F(x_{t+1}|x_t;\theta) - \nabla_{x_t} \log \tilde{P}_B(x_t|x_{t+1};\theta_B) = -\nabla_{x_t} \log \tilde{F}(x_t;\phi) \tag{52}$$

Therefore, we have the general *residual* $\nabla$-DB losses:

$$L_{\text{forward, general}}(x_t, x_{t+1})$$
$$= \left\| \nabla_{x_{t+1}} \log \tilde{P}_F(x_{t+1}|x_t;\theta) - \nabla_{x_{t+1}} \log \tilde{P}_B(x_t|x_{t+1};\theta_B) - \nabla_{x_{t+1}} \log \tilde{F}(x_{t+1};\phi) \right\|^2, \tag{53}$$

and

$$L_{\text{reverse, general}}(x_t, x_{t+1})$$
$$= \left\| \nabla_{x_t} \log \tilde{P}_F(x_{t+1}|x_t;\theta) - \nabla_{x_t} \log \tilde{P}_B(x_t|x_{t+1};\theta_B) + \nabla_{x_t} \log \tilde{F}(x_t;\phi) \right\|^2. \tag{54}$$

## D  COMMONS AND DIFFERENCES BETWEEN GFLOWNET AND SOFT RL

### D.1  GFLOWNET IN THE LENS OF SOFT RL

We restate the finding [64, 7] that, by constructing a special MDP, we may prove that GFlowNet and soft RL are equivalent when $P_B$ is fixed.

**Theorem 7.** *Let $\mathcal{M} = (\mathcal{G}, r)$ be an MDP define on a DAG $\mathcal{G}$ and a reward function $r(s, s')$ (defined for any transition $s \to s'$ on $\mathcal{G}$). Let $P_B$ be an arbitrary backward transition probability on $\mathcal{G}$. Suppose for any complete trajectory $\tau = (s_0, s_1, \ldots, s_T)$, we have the intermediate reward*

$$r(s_t, s_{t+1}) = \begin{cases} \frac{1}{\beta} \log P_B(s_t \mid s_{t+1}) & , t \neq T-1, \\ \log R(s_T) + \frac{1}{\beta} \log P_B(s_t \mid s_{t+1}) & , t = T-1. \end{cases}$$

*Then the optimal policy with respect to the following objective (where $\mathcal{H}$ is the entropy function)*

$$\pi^\star = \arg\max_\pi \mathbb{E}_{\tau \sim \pi} \left[ \sum_{t=0}^T r(s_t, s_{t+1}) + \frac{1}{\beta} \mathcal{H}\left(\pi(\cdot \mid s_t)\right) \right]$$

*samples terminating states $x$ with probability proportional to $R(x)^\beta$.*

**Remark 8.** The introduction of the intermediate reward terms $\log P_B(s_t|s_{t+1})$ (with fixed $P_B$) essentially anchors the optimal solution around the reference backward policy, which is rather natural from the generative model perspective and beneficial in creating better optimization landscapes.

**Remark 9.** Soft RL does not cover the case where $P_B$ can be jointly learned.

## D.2 DERIVATION OF $\nabla$-DB WITH SOFT Q-LEARNING

With soft Q-learning [15] (a special case of path consistency learning [42]) in the MDP defined in Sec D.1 with a fixed $P_B$, one aims at

$$\Delta_{\mathrm{SQL}}(s \to s') \triangleq Q(s, s') - (r(s, s') + V(s')) = 0 \tag{55}$$

where the soft value function is $V(s') \triangleq \frac{1}{\beta} \log \sum_{s':s \to s'} \exp(\beta Q(s', s''))$ and $Q(s, s')$ is the soft Q-value function. One can show that

$$P_F(s'|s) = \frac{\exp(\beta Q(s, s'))}{\sum_{s'} \exp(\beta Q(s, s'))}. \tag{56}$$

Therefore, at optimality where $\Delta_{\mathrm{SQL}}(s \to s') = 0$ for all $s \to s'$, we can plug in the definitions above and obtain

$$\log P_F(s'|s) - \log P_B(s|s') = V(s') - V(s) \tag{57}$$

which is in the same form of the DB condition. Taking gradients of the above condition with respect to both $s$ and $s'$ leads to equivalent $\nabla$-DB conditions.

## E CORRECTIONS FOR NON-IDEAL PRETRAINED MODELS

For non-ideal pretrained models in which $P_F^{\#}$ does not match $P_B$ (*i.e.*, the DB condition is violated), the original *residual* $\nabla$-DB objective is apparently biased. We may introduce an additional learnable term $h(x_t, x_{t+1}; \psi)$ to compensate for this error, with which we have the corrected DB condition for the pretrained model:

$$\log P_F^{\#}(x_{t+1}|x_t) - \log P_B(x_t|x_{t+1}) = \log F^{\#}(x_{t+1}) - \log F^{\#}(x_t) - h(x_t, x_{t+1}) \tag{58}$$

Hence, we have the following *residual* $\nabla$-DB losses in the case of the imperfect pretrained model:

$$L_{\overrightarrow{\nabla}\text{DB-res-v2}}(x_t, x_{t+1}) = \left\| \nabla_{x_{t+1}} \log \tilde{P}_F(x_{t+1}|x_t) - \nabla_{x_{t+1}} \log \tilde{F}(x_{t+1}) + \nabla_{x_{t+1}} h(x_t, x_{t+1}; \psi). \right\|^2, \tag{59}$$

$$L_{\overleftarrow{\nabla}\text{DB-res-v2}}(x_t, x_{t+1}) = \left\| \nabla_{x_t} \log \tilde{P}_F(x_{t+1}|x_t) + \nabla_{x_t} \log \tilde{F}(x_{t+1}) + \nabla_{x_t} h(x_t, x_{t+1}; \psi). \right\|^2, \tag{60}$$

and the bidirectional $\nabla$-DB loss for learning $h$ in the pretrained model:

$$\begin{aligned} L_{\overrightarrow{\nabla}\text{DB-pretrained}}(x_t, x_{t+1}, x_{t+2}) = \Big\| &\nabla_{x_{t+1}} \log P_F^{\#}(x_{t+1}|x_t) + \nabla_{x_{t+1}} \log P_F^{\#}(x_{t+2}|x_{t+1}) \\ &- \nabla_{x_{t+1}} \log P_B(x_t|x_{t+1}) - \nabla_{x_{t+1}} \log P_B(x_{t+1}|x_{t+2}) \\ &+ \nabla_{x_{t+1}} h(x_t, x_{t+1}; \psi) + \nabla_{x_{t+1}} h(x_{t+1}, x_{t+2}; \psi) \Big\|^2. \end{aligned} \tag{61}$$

## F  MDP CONSTRUCTION FOR DIFFUSION MODELS

Many typical inference algorithms (or samplers) of diffusion models, including but not limited to DDPM [17], DDIM [59] and SDE-DPM-solver++ [38], can be abstracted into a loop with the following two steps (with the convention of GFlowNets on time indexing):

1. **Computation of predicted clean data.** $\hat{x}_T = f(x_t, t)$.
2. **Back-projection of the predicted clean data.** $x_{t+1} \sim \mathcal{N}(g(\hat{x}_T, t+1), \sigma_{t+1}I)$

The MDP for diffusion models can therefore be simply constructed as:

- State: $(x_t, t)$.
- Transition: $x_t \sim \mathcal{N}\Big(g(f(x_{t-1}, t-1), t), \sigma_t I\Big)$.
- Starting state: $x_0 \sim P(x_0) = \mathcal{N}(0, \sigma_0 I)$.
- Terminal state: $(x_T, T)$.
- Terminal reward: $R(x_T, T)$.
- Intermediate reward: $R(x_t, t) = 0$.

Since the intermediate rewards are always zero, for terminal rewards we simply write $R(x_T)$ without the second argument of $T$.

With this MDP defined, it is straightforward to apply policy gradient methods to optimize the return of the sampled on-policy trajectories. For instance, DDPO employs PPO [55] to perform updates.

## G  APPLICATION TO FLOW MATCHING MODELS

Flow matching models [31] are a popular and powerful class of generative models that sample points via simulation but of an ordinary differential equation (ODE) instead of an SDE. Specifically, a flow matching model defines a velocity field $v(x, t)$ and, by starting from a randomly initialized $x(0) = x_0 \sim \mathcal{N}(0, I)$, generate samples $x_1 = x(1)$ with $\dot{x} = v(x, t)$. Since this is a deterministic process, our $\nabla$-GFlowNet does not naïvely apply.

However, it is shown that a flow matching model above can be turned into an equivalent family of SDEs with the same probability flow, with an arbitrary diffusion term $\sigma(t)$ [10]:

$$
\mathrm{d}X_t = \left( v(X_t, t) + \frac{\sigma(t)^2}{2\beta_t \left( \frac{\dot{\alpha}_t}{\alpha_t} \beta_t - \dot{\beta}_t \right)} \left( v(X_t, t) - \frac{\dot{\alpha}_t}{\alpha_t} X_t \right) \right) \mathrm{d}t + \sigma(t)\, \mathrm{d}B_t, \quad X_0 \sim \mathcal{N}(0, I).
\tag{62}
$$

This is essentially a diffusion model but with a non-linear noising process. Since we can always fix this noising process (*i.e.*, $P_B$ in our setting) during finetuning, we can use $\nabla$-GFlowNet to obtain a finetuned diffusion model from this base diffusion model.

## H    FINETUNING CONVERGENCE IN WALL TIME

We further show the convergence speed measured in relative wall time on a single node with 8 80GB-mem A100 GPUs in Fig. 9.

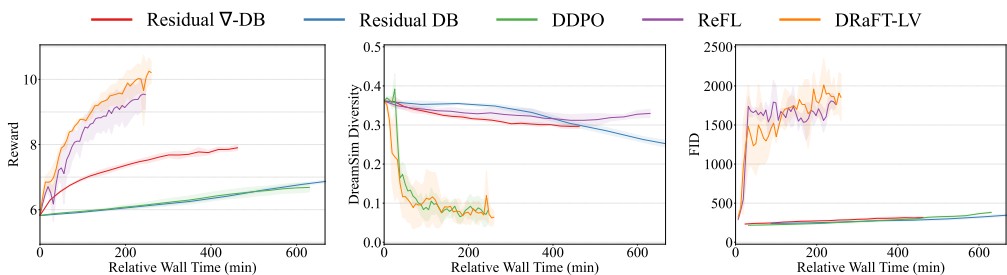

Figure 9: Convergence curves of different metrics for different methods throughout the finetuning process on Aesthetic Score, with the $x$-axis being the relative wall time. All methods are benchmarked on a single node with 8 80GB-mem A100 GPUs.

# I RESULTS OF ABLATION STUDY

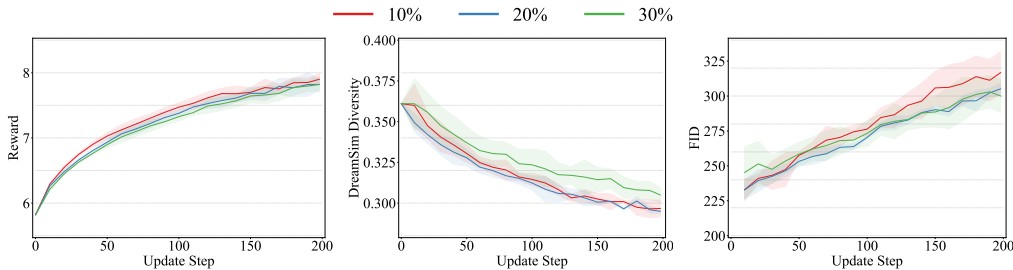

Figure 10: Ablation study on the effect of subsampling rate on the collected trajectories for computing the *residual* ∇-DB loss.

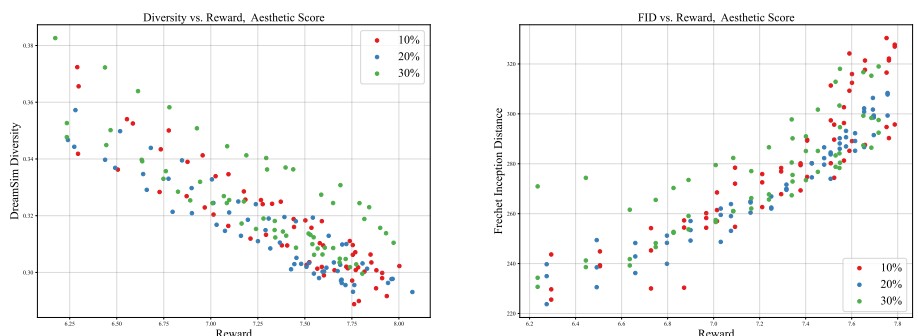

Figure 11: Pareto frontiers for reward, diversity and prior-preservation (measured by FID) of models trained with different subsampling rate. In expectation, higher subsampling rates seem to slightly help in increasing diversity.

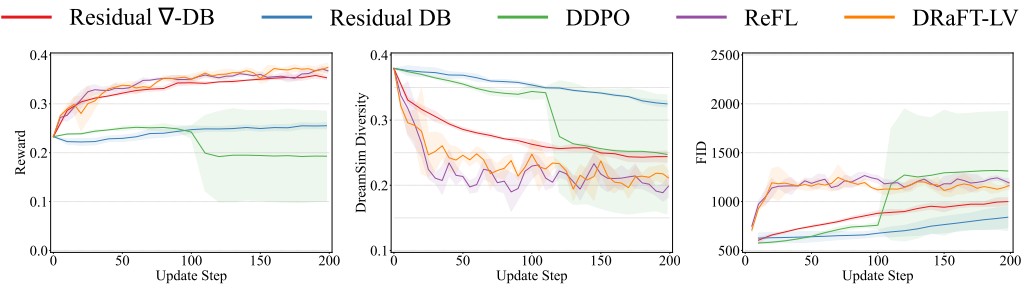

Figure 12: Convergence curve of metrics of different methods throughout the finetuning process on the HPSv2 reward model.

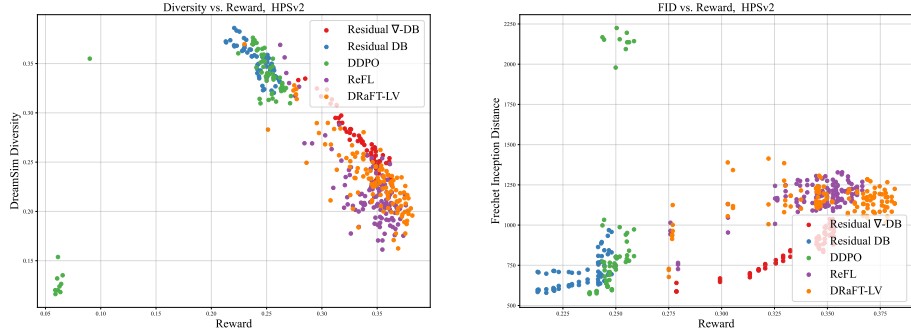

Figure 13: Pareto frontiers for reward, diversity and prior-preservation (measured by FID) on HPSv2.

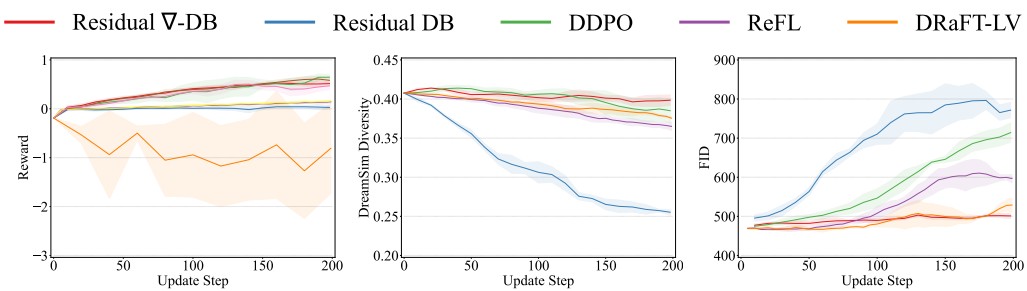

Figure 14: Convergence curve of metrics of different methods throughout the finetuning process on the ImageReward reward model.

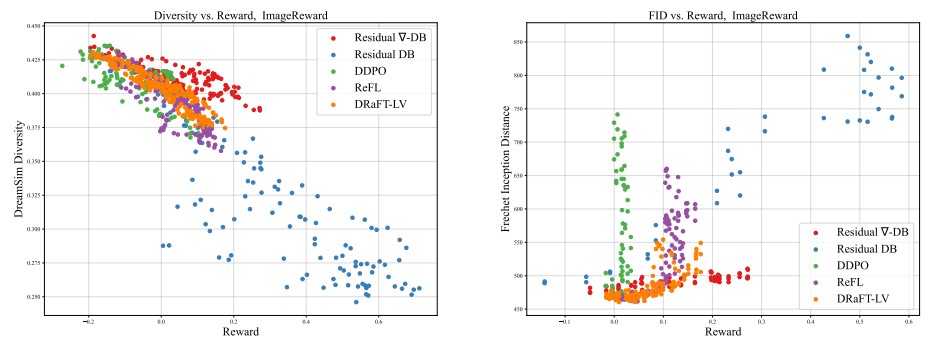

Figure 15: Pareto frontiers for reward, diversity and prior-preservation (measured by FID) on ImageReward.

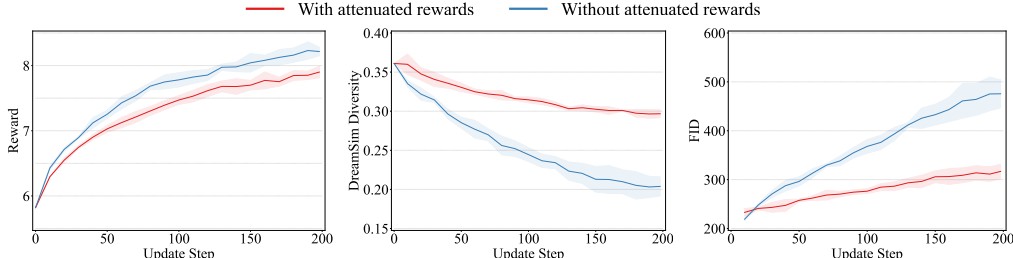

Figure 16: Convergence curve of metrics of different methods throughout the finetuning process on Aesthetic Score with time-dependent attenuation of predicted rewards. Both models are trained with $\beta = 10000$. With decayed predicted rewards, the convergence speed is slower but due to less aggressive prediction on reward signal, the model with reward attenuation achives better diversity and prior-preserving results.

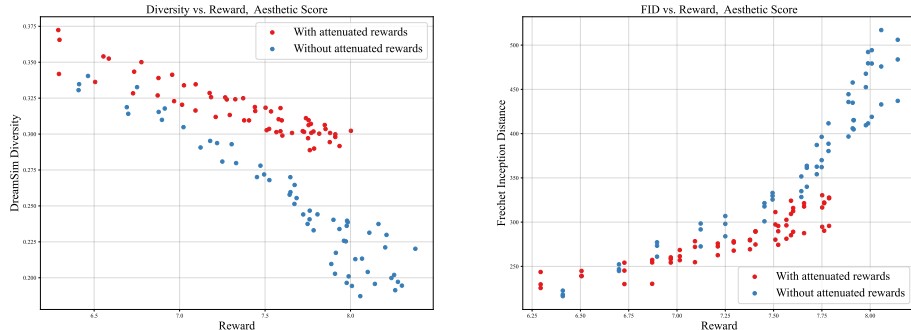

Figure 17: Pareto frontiers for reward, diversity and prior-preservation (measured by FID) on Aesthetic Score with time-dependent scaling of predicted rewards.

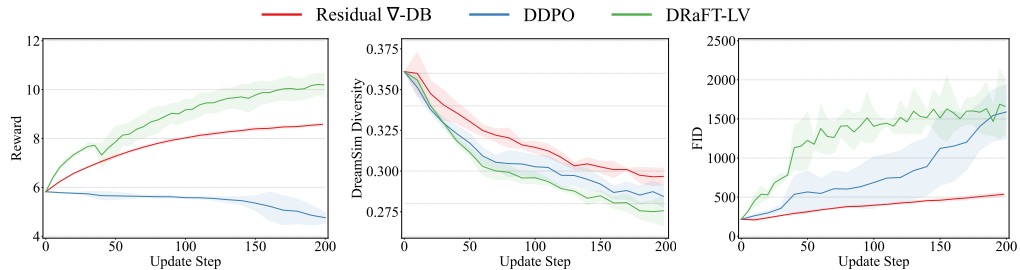

Figure 18: Convergence curve of metrics of different methods throughout the finetuning process on Aesthetic Score with the MDP constructed by SDE-DPM-Solver++ (with 20 inference steps).

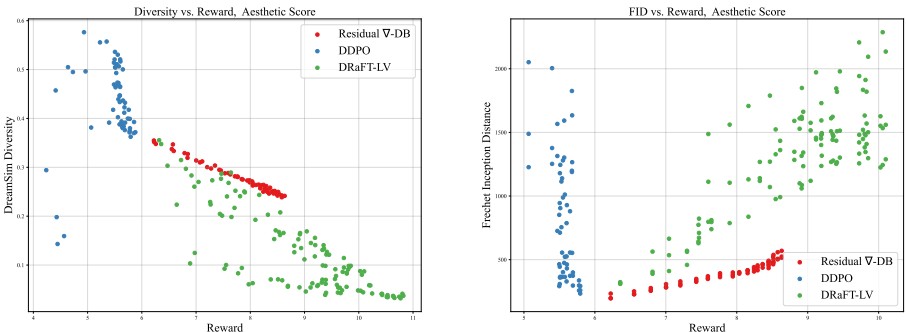

Figure 19: Pareto frontiers for reward, diversity and prior-preservation (measured by FID) on Aesthetic Score with the MDP constructed by SDE-DPM-Solver++ (with 20 inference steps).

## J  ADDITIONAL ABLATION EXPERIMENTS

**Effect of different prior strengths.** It is often of interest to have slightly weaker prior instead of simply increasing the reward strength, of which the sampling objective is $R(x_T)^\beta P_F^{\#}(x_T)^\eta$. While it is hard to obtain an exact estimate of $P_F^{\#}(x_T)^\eta$ where $\eta \in (0,1]$ is the prior strength, one can approximate it with the weighted score function $\eta \nabla_{x_t} \log P_F^{\#}(x_t, t)$. Denoting $F_\eta(x_t)$ the corresponding flow, we have the modified *residual* $\nabla$-DB conditions.

$$\underbrace{\nabla_{x_{t+1}} \log P_F(x_{t+1}|x_t) - \eta \nabla_{x_{t+1}} \log P_F^{\#}(x_{t+1}|x_t)}_{\nabla_{x_{t+1}} \log \tilde{P}_F(x_{t+1}|x_t):\text{ residual policy score function}} = \underbrace{\nabla_{x_{t+1}} \log F(x_{t+1}) - \nabla_{x_{t+1}} \log F_\eta^{\#}(x_{t+1})}_{\nabla_{x_{t+1}} \log \tilde{F}(x_{t+1}):\text{ residual flow score function}}.$$

(63)

$$\underbrace{\nabla_{x_t} \log P_F(x_{t+1}|x_t) - \eta \nabla_{x_t} \log P_F^{\#}(x_{t+1}|x_t)}_{\nabla_{x_t} \log \tilde{P}_F(x_{t+1}|x_t):\text{ reverse residual policy score function}} = \underbrace{\nabla_{x_t} \log F(x_t) - \nabla_{x_t} \log F_\eta^{\#}(x_{t+1})}_{\nabla_{x_t} \log \tilde{F}(x_t):\text{ residual flow score function}}.$$

(64)

We experiment with choices of prior strengths $\eta$ and observed in Fig. 20 and 21 that lower $\eta$ lead to better diversity-reward trade-off and faster reward convergence.

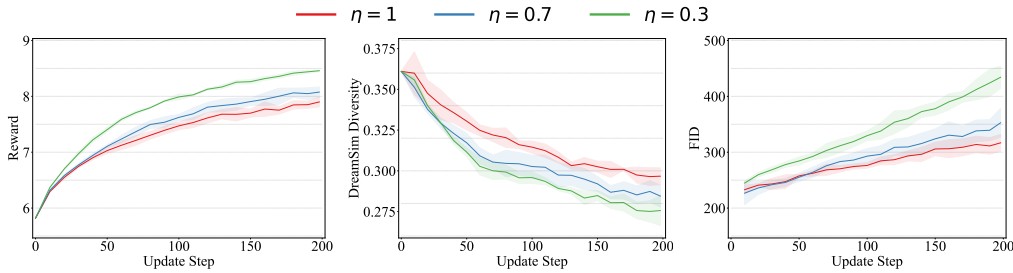

Figure 20: Convergence curve of metrics of different methods throughout the finetuning process on Aesthetic Score with different prior strength $\eta$.

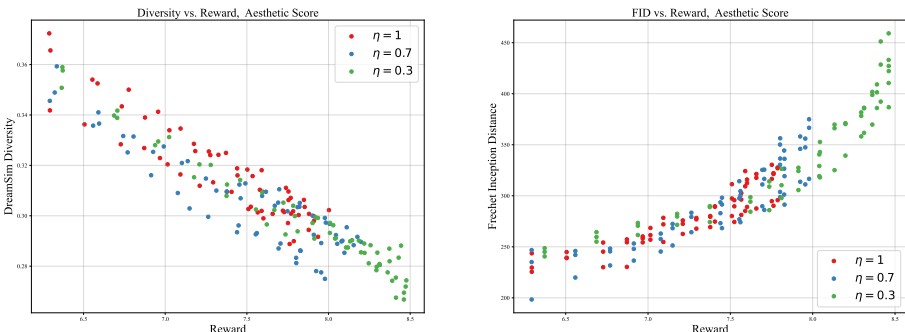

Figure 21: Pareto frontiers for reward, diversity and prior-preservation (measured by FID) on Aesthetic Score with different prior strength $\eta$.

**Effect of 2nd-order gradients in finetuning.** In Fig. 22 and 23, we show the comparison between models with and without 2nd-order gradients, where both models are trained with $\beta = 10000$. Empirically, 2nd-order gradients hurts the trade-off between reward convergence, diversity preservation and prior preservation.

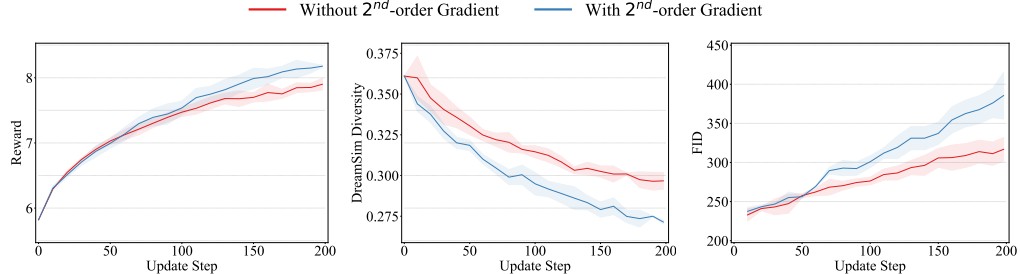

Figure 22: Convergence curve of metrics of different methods throughout the finetuning process on Aesthetic Score with and without 2nd-order gradients.

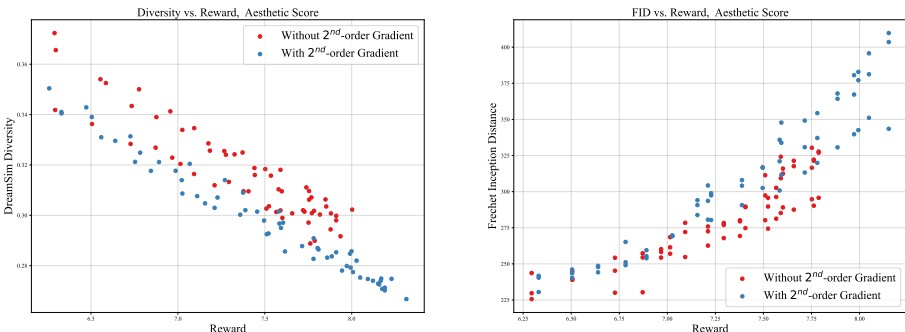

Figure 23: Pareto frontiers for reward, diversity and prior-preservation (measured by FID) on Aesthetic Score with and without 2nd-order gradients.

# K  MORE SAMPLES (AESTHETIC SCORE)

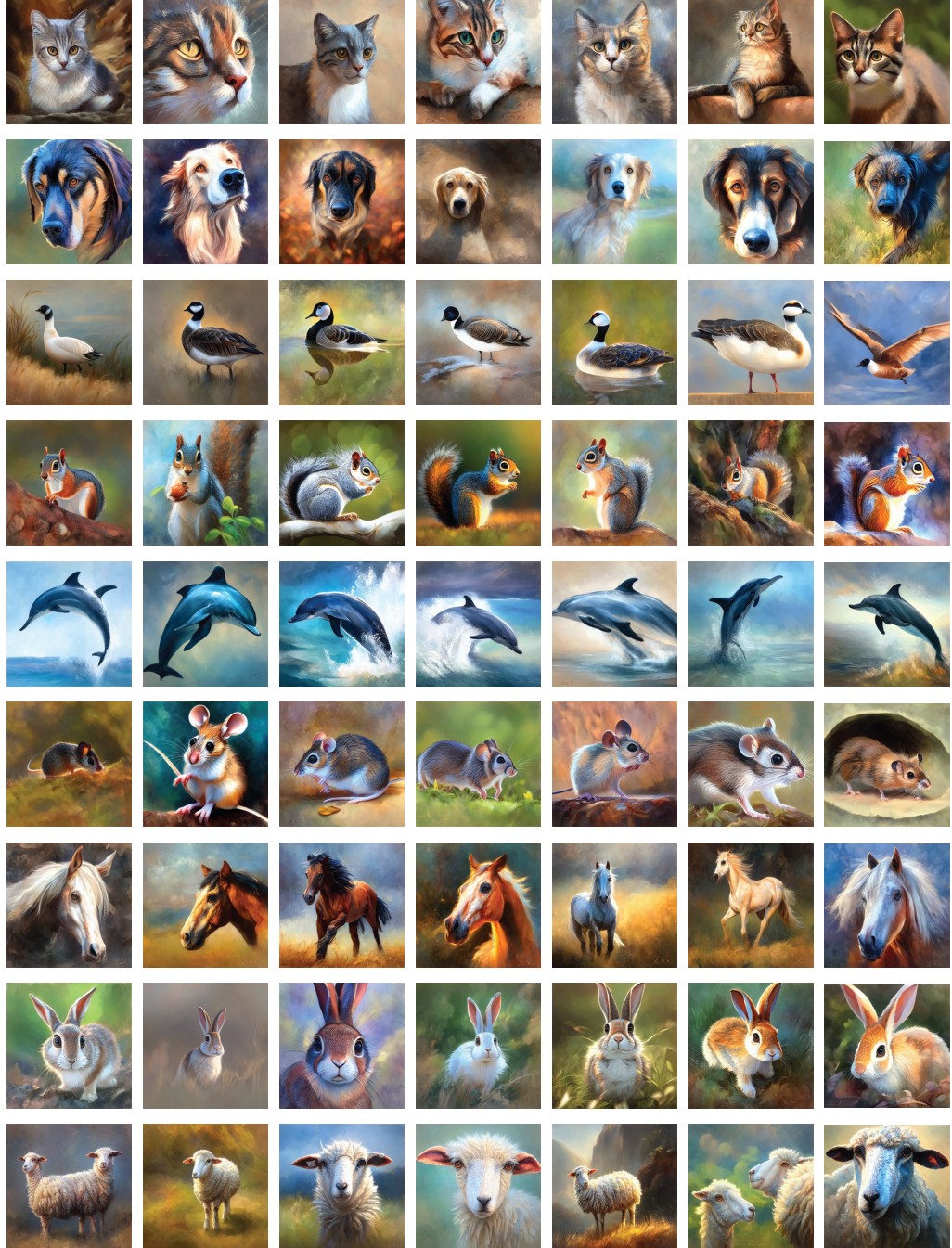

Figure 24: Additional uncurated samples from the model finetuned with *residual* ∇-DB on the reward model of Aesthetic Score.

## L   MORE SAMPLES (HPSv2)

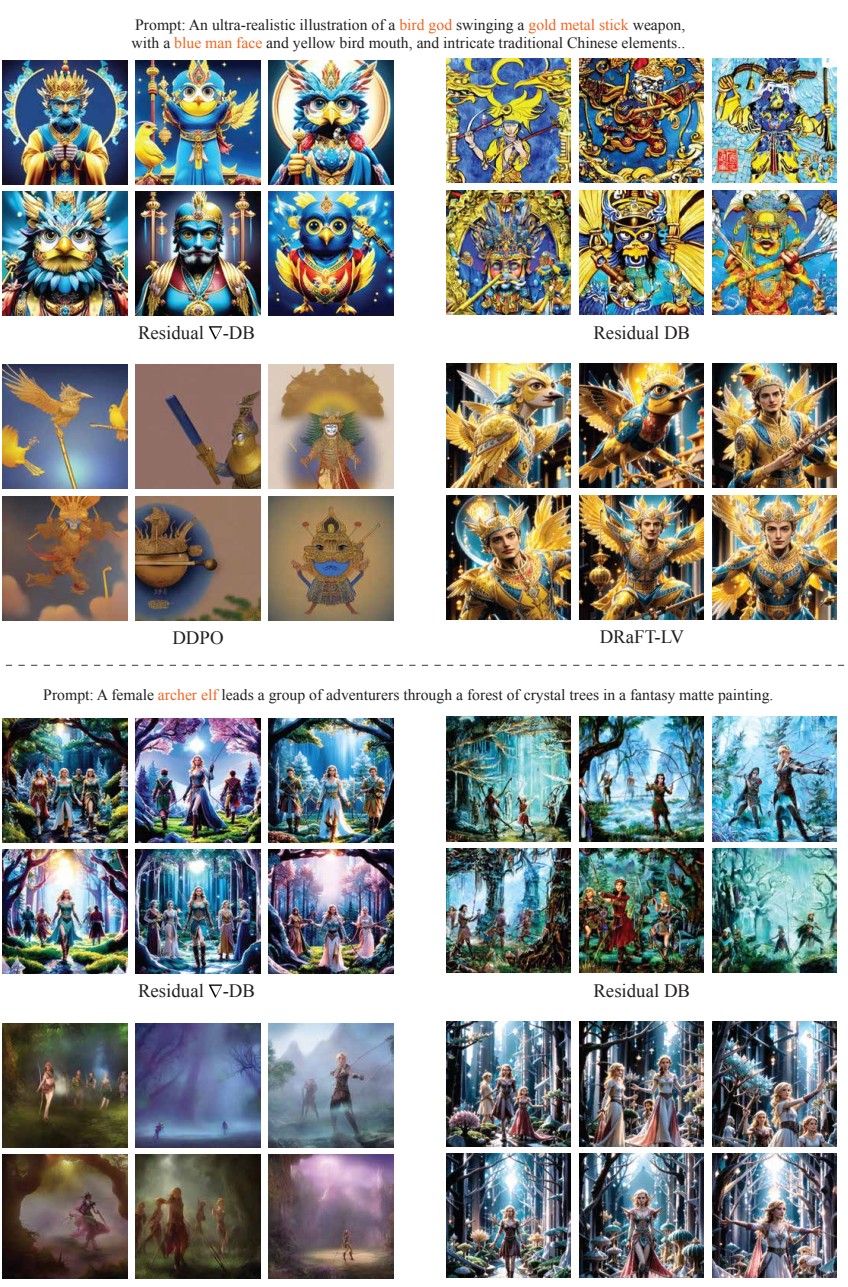

Figure 25: More comparison between samples generated by *residual* ∇-DB and the baseline methods. The model finetuned with *residual* ∇-DB is capable of following the instructions while generating diverse samples.

Prompt: A surreal cat with a smile and intricate details.

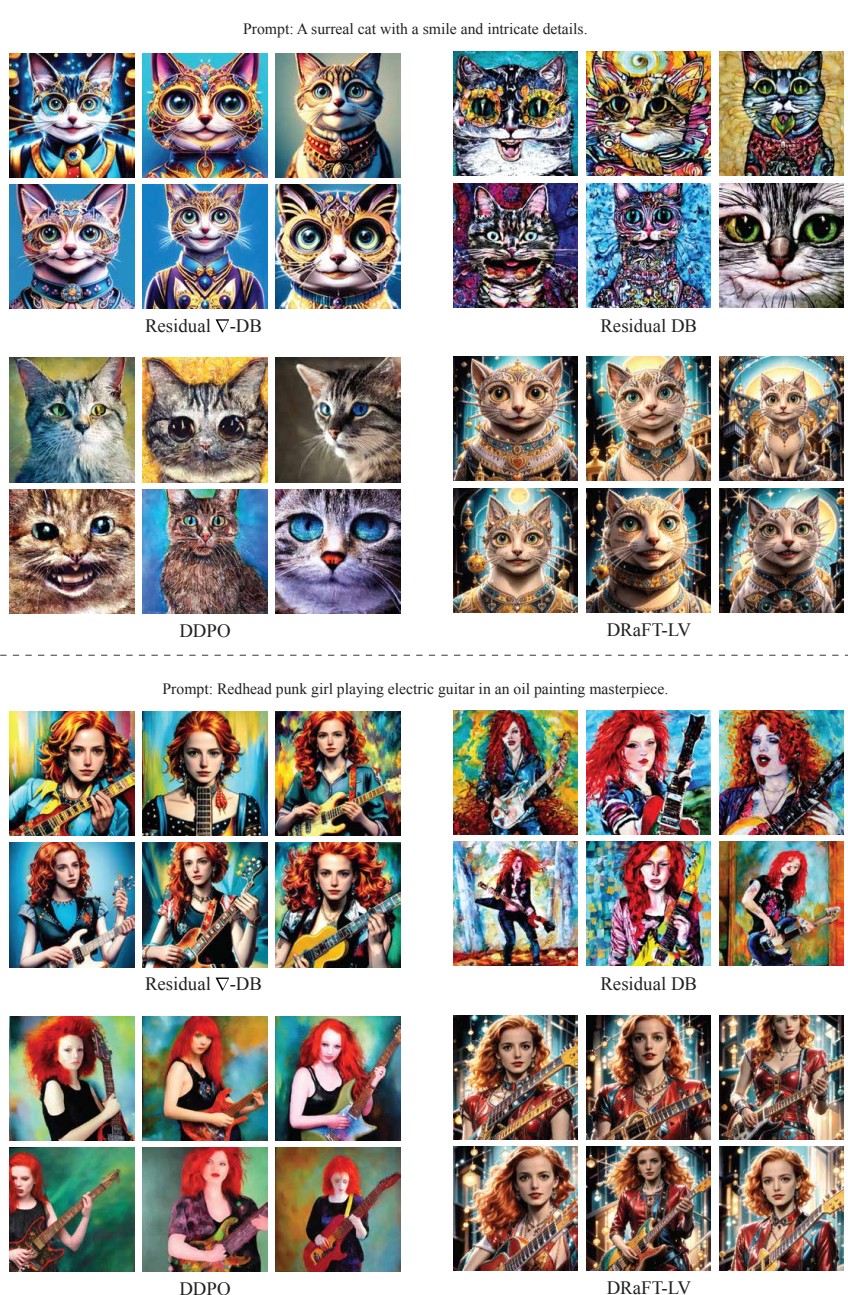

Figure 26: HPSv2 samples, Continued.

# M  QUALITATIVE COMPARISON ON DIVERSITY OF SAMPLES GENERATED BY DIFFERENT METHODS

Below we show uncurated samples generated by models finetuned with different methods with propmts *cat, bird, rabbit, bird, kangaroo*.

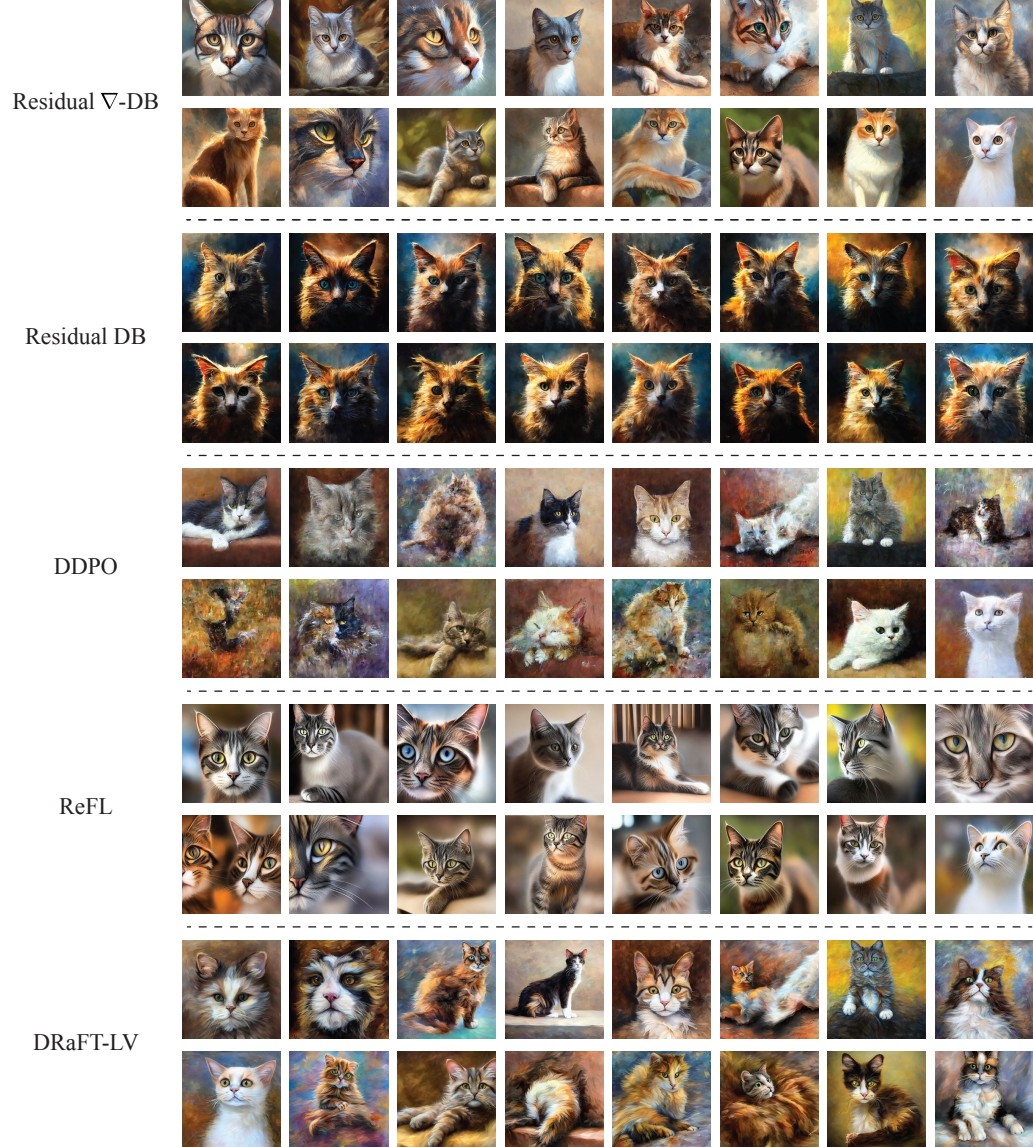

Figure 27: Uncurated samples generated by models finetuned with different methods with prompt *cat*.

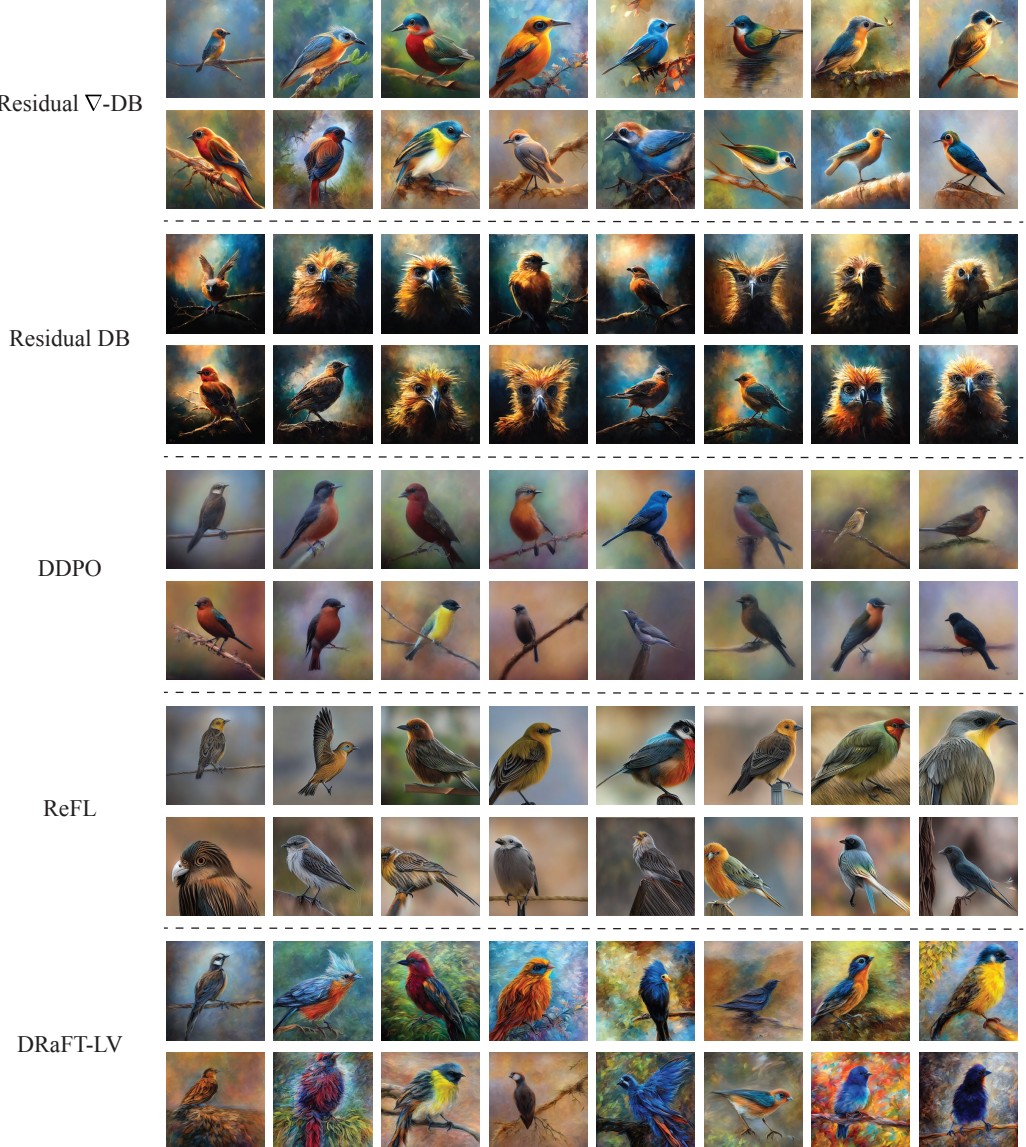

Figure 28: Uncurated samples generated by models finetuned with different methods with prompt *bird*.

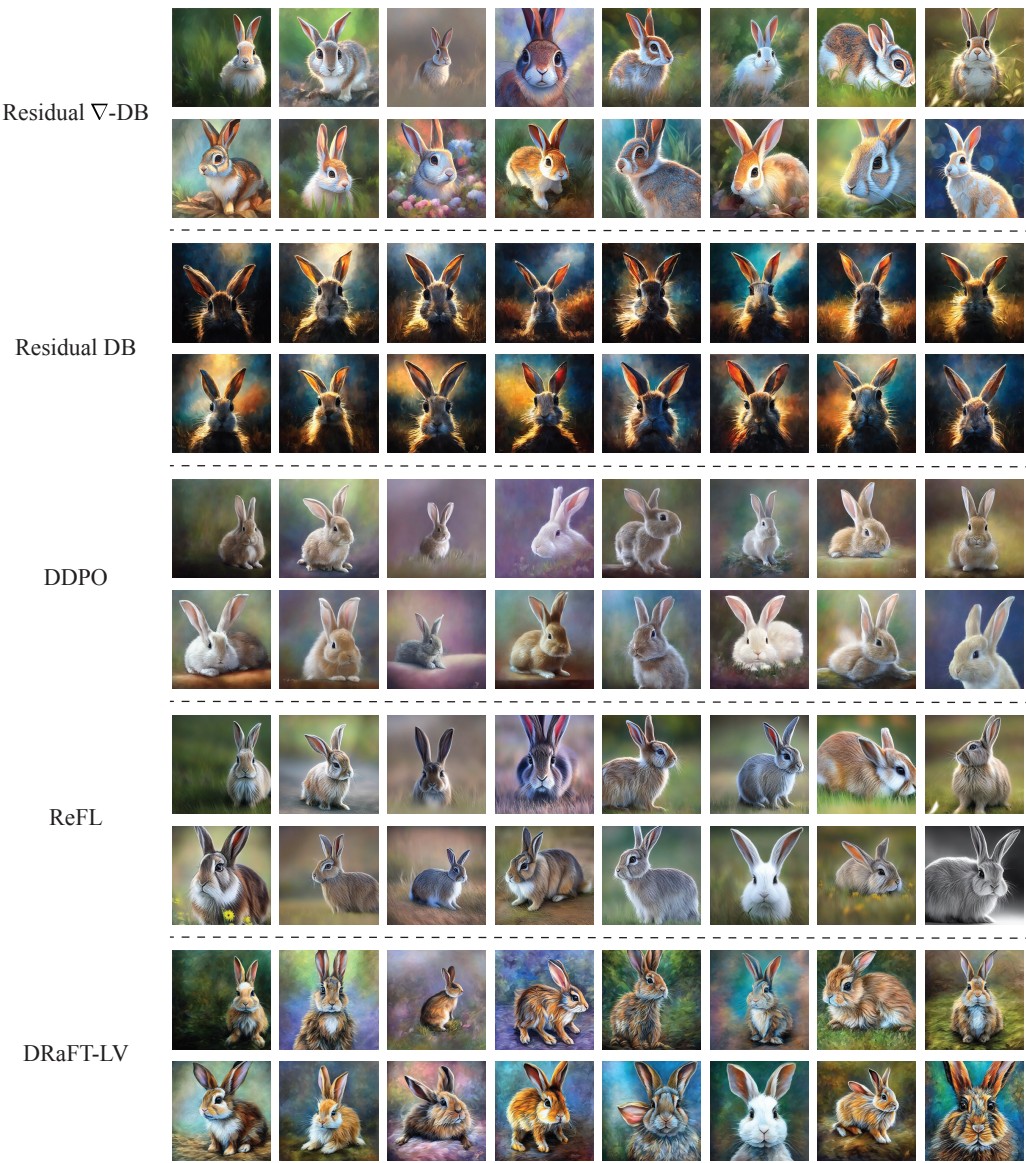

Figure 29: Uncurated samples generated by models finetuned with different methods with prompt *rabbit*.

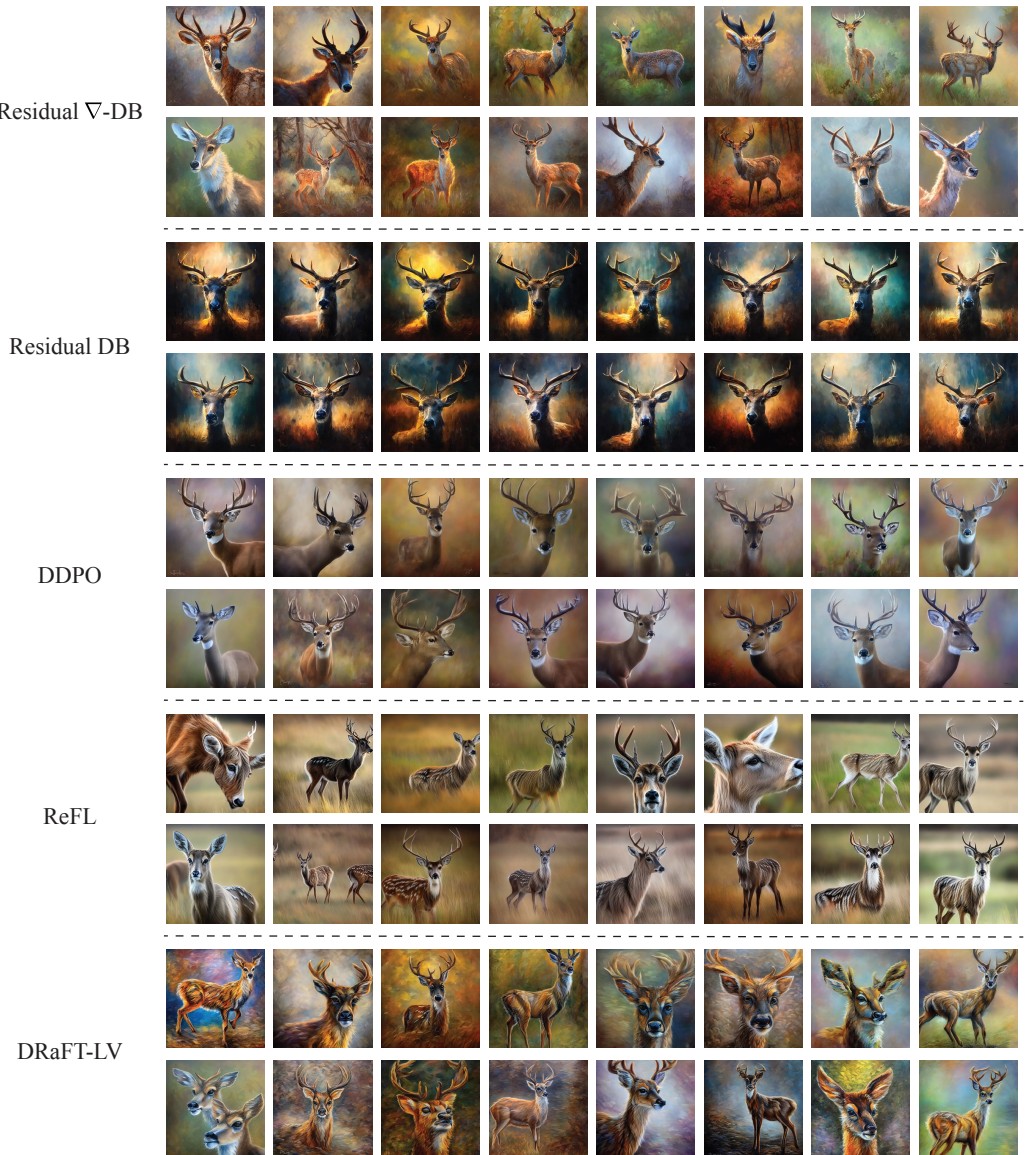

Figure 30: Uncurated samples generated by models finetuned with different methods with prompt *deer*.

Residual ∇-DB

Residual DB

DDPO

ReFL

DRaFT-LV

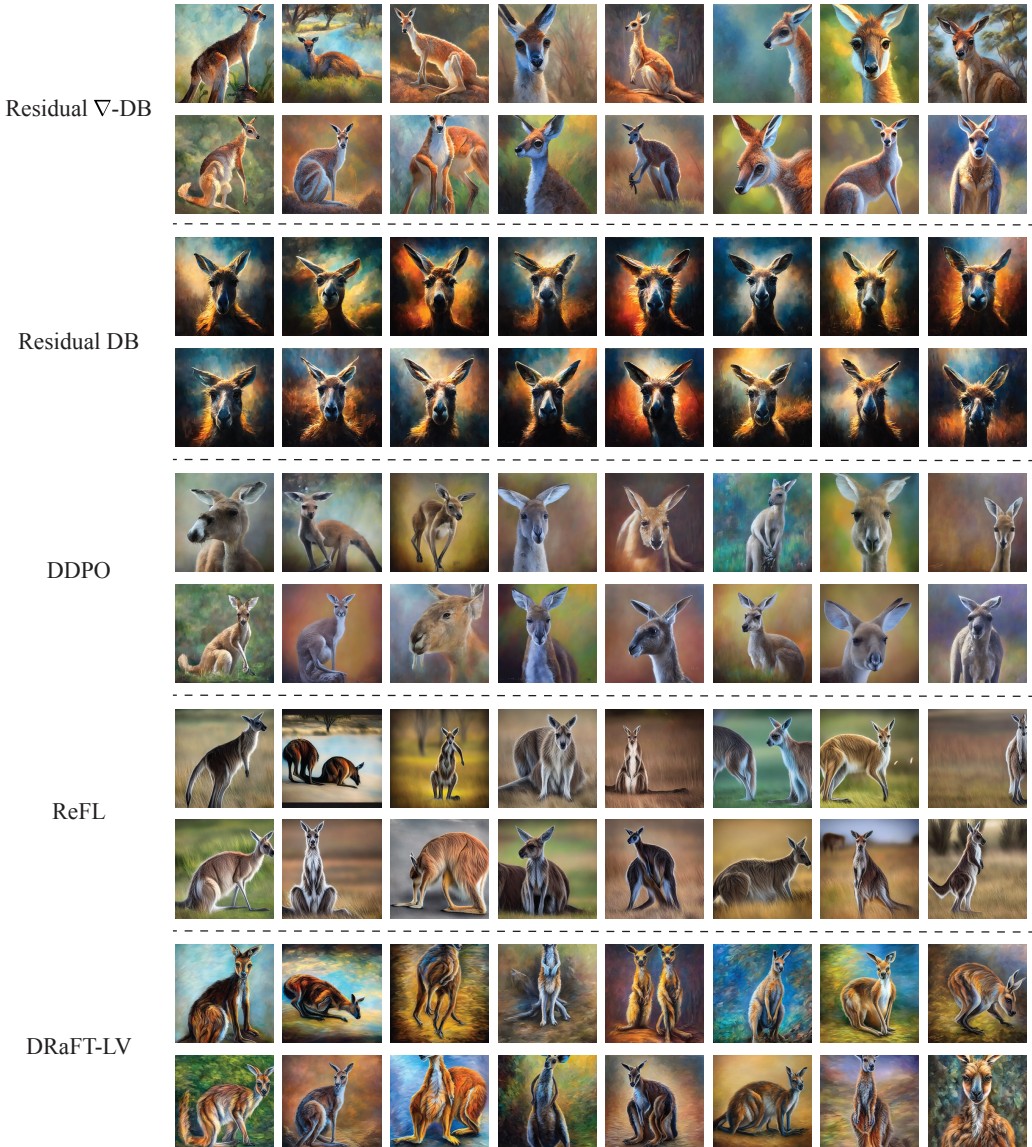

Figure 31: Uncurated samples generated by models finetuned with different methods with prompt *kangaroo*.

