# OpenReview forum: "Efficient Diversity-Preserving Diffusion Alignment via Gradient-Informed GFlowNets"
_ICLR.cc/2025/Conference — ICLR 2025 Poster_

### Official Review · Reviewer_5qSt · 2024-11-01

**Soundness:** 3
**Presentation:** 2
**Contribution:** 2
**Rating:** 6
**Confidence:** 2

**Summary:**

This paper introduces Nabla-GFlowNet with a new objective, $ \nabla$-DB, and its variant, residual $ \nabla$-DB, for finetuning pretrained diffusion models. The method aims to enhance sample diversity and finetuning efficiency by utilizing reward gradients. Experiments on two reward functions, Aesthetic Score and Human Preference Score (HPSv2), demonstrate improved performance.

**Strengths:**

- The paper has a clear motivation, addressing the challenge of preserving diversity and improving efficiency in finetuning diffusion models, which is crucial for real-world applications.
- The paper introduces a unique application of GFlowNet principles to diffusion model finetuning, specifically focusing on the reward gradient to preserve sample diversity.

**Weaknesses:**

- In the quantitative evaluation, the proposed method performs well on the smaller dataset of Aesthetic Score, nearly doubling the baseline in the DreamSim metric. However, on the larger dataset of HPSv2, its performance is similar to baselines, showing no clear advantage. The effectiveness of the method remains inconclusive due to the differences in dataset size.
- While the authors mention “fast and efficient finetuning” as a contribution, only Figure 4 shows comparable convergence speed to the baseline. It would be helpful to include details on training resource consumption, such as GPU usage and computational cost, to substantiate this claim.
- Figure 3 only compares the results of the pretrained model and the proposed method, lacking visual comparisons with other baselines.
- Figure 5 lacks sufficient information in the title and annotations to clarify what each point represents (e.g., training iteration). To improve clarity, consider adding an explanation in the figure caption or in Section 4.4 to specify the meaning of each point.
- The conclusion section is missing, which may limit the clarity of the paper’s overall findings and contributions.

**Questions:**

Please see weaknesses.

---

> ### Author Response · Authors · 2024-11-21
>
> We thank the reviewer for their time and efforts and their constructive comments. We have made some further refinements on the paper presentation and the experiments (detailed in the general response).
>
> >  In the quantitative evaluation, the proposed method performs well on the smaller dataset of Aesthetic Score, nearly doubling the baseline in the DreamSim metric. However, on the larger dataset of HPSv2, its performance is similar to baselines, showing no clear advantage. The effectiveness of the method remains inconclusive due to the differences in dataset size.
>
> Thanks for raising this concern. In our revised draft, we show that on HPSv2 our method achieves the best performance (comparable speed to ReFL, yet still with good diversity and prior preservation), after we 1) introduce the time-dependent scaling of the predicted reward and 2) fix some numerical issues during mixed-precision training (Table 1 and Figure 12 & 13).
>
> In the meantime, we would like to point out that the evaluation results in the presented table is probably misleading. A method can converge really fast (in terms of reward) but fail to output diverse outputs or fail to follow the pretrained prior (in the worst case, producing nonsensical outputs, as illustrated in Fig 3 in the revised draft). Therefore, we believe that the best way to evaluate these methods is to look at the “Pareto frontiers” (quite similar to the precision-recall curve), where one can see the performance of different checkpoints of the same model and directly compare both diversity- and prior-preservation capability of model checkpoints with similar rewards.
>
> On the dataset/reward model concern: the reward model of Aesthetic Score is indeed trained on a relatively large dataset (LAION-aesthetic, a subset with ~238k images from the LAION dataset [1]) whereas HPSv2 is trained on HPDv2 dataset [2] of ~433k pairs of images. The major differences between these reward models are more related to the specific objectives. Specifically, Aesthetic Score cares more about the style of images (which is less correlated with text prompts), while HPSv2 demands more on alignment between text prompts and generated images.
>
> > While the authors mention “fast and efficient finetuning” as a contribution, only Figure 4 shows comparable convergence speed to the baseline. It would be helpful to include details on training resource consumption, such as GPU usage and computational cost, to substantiate this claim.
>
> Good point! We now include the convergence plots (Figure 9) with the x-axis being the relative wall time to quantitatively evaluate how costly (in time) each method can be, with the caption stating the amount of compute used.
>
> > Figure 3 only compares the results of the pretrained model and the proposed method, lacking visual comparisons with other baselines.
>
> Thank you for pointing this issue out.  We have included more qualitative comparisons based on your suggestions, in both the experiment section (Figure 4 and 5 in the revised draft) and the last two pages of the appendix (Figure 24, 25 & 26).
>
> > Figure 5 lacks sufficient information in the title and annotations to clarify what each point represents (e.g., training iteration). To improve clarity, consider adding an explanation in the figure caption or in Section 4.4 to specify the meaning of each point.
>
> Thank you for reminding us of our overlook on explaining the Pareto frontier figure. We have revised the caption (Figure 7 in the revised draft) and hopefully it is clearer now.
>
> > The conclusion section is missing, which may limit the clarity of the paper’s overall findings and contributions.
>
> Thanks for your suggestion. We now include a conclusion section that summarizes the contributions.
>
> ---
> [1] https://laion.ai/blog/laion-aesthetics/
>
> [2] Human Preference Score v2: A Solid Benchmark for Evaluating Human Preferences of Text-to-Image Synthesis. Xiaoshi Wu, Yiming Hao, Keqiang Sun, Yixiong Chen, Feng Zhu, Rui Zhao, Hongsheng Li. https://arxiv.org/abs/2306.09341

---

> > ### Comment · Reviewer_5qSt · 2024-11-26
> >
> > Thank you for your detailed response. I appreciate the clarifications and the efforts you've made to address most of my concerns. I have decided to raise my rating to 6.

---

> > > ### Author Response · Authors · 2024-12-03
> > >
> > > As it is coming to the end of the rebuttal period, we would like to deeply appreciate the reviewer's time and efforts in positively re-evaluating our paper, and we are happy to see that we addressed most of your concerns.

---

### Official Review · Reviewer_5N8Z · 2024-11-03

**Soundness:** 3
**Presentation:** 4
**Contribution:** 3
**Rating:** 6
**Confidence:** 3

**Summary:**

The authors propose a method, called $\nabla$-GFlowNet which which is modification of GFLowNets. They define a new objective $\nabla$-DB, which is a gradient informed version the Detailed Balance objective. The paper then goes on to propose a residual version of this loss which at optimality samples proportionally to the argumented distribution $r(x_T)p^\sharp (x_T)$, which maintains diversity of generations. The paper then presents experiments which shows a good pareto frontier of reward vs diversity on two tasks.

**Strengths:**

- This paper offers a good way at maintaining diversity in generations while aligning generations to a reward model.
- The paper is theoretically founded and shows that their new training objective maintains the validity of GFlowNets while taking into account gradient information.
- The work has a number of good ablations and experiments to show diversity and reward tradeoff.

**Weaknesses:**

- The problem of bias is well known in optimizations for diffusion models (elaborated in question section). Some treatment of this problem problem would be desirable since it seems that this method may be optimizing for the biased distribution

**Questions:**

- It is known that optimizing the KL constrained optimization problem, with the closed form solution of the augmented distribution, can lead to a biased result [0, 1]. Does this method have this bias problem? If not, how do you get around it?

[0] https://arxiv.org/abs/2409.08861
[1] https://arxiv.org/abs/2402.15194

---

> ### Author Response · Authors · 2024-11-21
>
> We thank the reviewer for their acknowledge on our paper's contribution and sharing their concerns. We have made some further refinements on the paper presentation and the experiments (detailed in the general response).
>
> > The problem of bias is well known in optimizations for diffusion models (elaborated in question section). Some treatment of this problem problem would be desirable since it seems that this method may be optimizing for the biased distribution
>
> Thank you for raising this interesting point, which indeed demonstrates the advantage of our method.
>
> The source of the bias in the papers cited above lies in their **stochastic optimal control** (SOC) formulation. As shown in Equation 19 in the adjoint matching paper [0], the “reward” in the equivalent RL formulation is $-\int_0^1 f(x_t, t)dt - g(x_1)$ on the terminal state $x_1$, which is different from $R(x_1)$ (if it were some RL method). It therefore inevitably depends on the initial value $V(x_0)$, as shown in Equation 20 and 23 in the paper. We note that this is different from the standard RL setting where one optimizes $R(x_T)$ which does not have this bias issue.
>
> In contrast, GFlowNets are proposed [2] in the first place to sample from non-negative rewards (i.e., unnormalized density functions) **in an unbiased way**. By directly optimizing the detailed balance (DB) condition (or the gradient-based version), the log-flow function (or the gradient of that) learns to correct any deviation from the target distribution. Therefore, instead of saying that we “get around” the bias issue, we would prefer to say that our framework is theoretically bias-free by design.
>
> ---
> [2] GFlowNet Foundations. Yoshua Bengio, Salem Lahlou, Tristan Deleu, Edward J. Hu, Mo Tiwari, Emmanuel Bengio. https://arxiv.org/abs/2111.09266

---

> > ### Comment · Reviewer_5N8Z · 2024-11-25
> > **Response to Official Comment**
> >
> > Thank you for the reply. I think this is correct (although not because they are optimizing a different function as mentioned, but because they add a second term to the drift coefficient). I would like to see some comparison to these SOC methods.
> >
> > In the absence of this, I will maintain my score since I believe that this is a strong paper, but still with limited impact.

---

> ### Author Response · Authors · 2024-11-30
>
> We sincerely thank the reviewer for their response and acknowledgement that our paper is a strong one.
>
> On your comment on the root of bias: yes, you are absolutely correct. Equation 20 and 23 in their paper are merely trying to provide equivalent RL objectives to make direct comparisons between their method and RL.
>
> ---
> We greatly appreciate that the reviewer pointed out SOC methods as baselines, which are indeed important and inspiring in the field of reward finetuning for diffusion models. And we would like to further share our opinions and results regarding your comments on comparison with SOC methods:
>
> **ELEGANT.**
>
> Directly comparison against ELEGANT is tricky, as ELEGANT essentially employs a 3-stage training procedure in which only the first one queries reward functions. Here we directly compare the final generated images with ELEGANT (directly copied and pasted from their paper) and our $\nabla$-GFlowNet in this anonymous link: [https://drive.google.com/file/d/1FLI1rypZCIg2qerVaa8dQNqq-3vNeiB7/view?usp=sharing](https://drive.google.com/file/d/1FLI1rypZCIg2qerVaa8dQNqq-3vNeiB7/view?usp=sharing). We may obverse that the generated images of ELEGANT suffer from collapse in generation, illustrated by their consistent production of distorted shapes of animals and a very consistent style of non-realistic images. In contrast, the generated images with our method show much better aesthetics (colorful images), diversity (as the subjects, animal poses and image styles vary) and prior preservation (as the images look much more realistic).
>
> **Adjoint matching.**
>
> - It is a really new paper, of which the first arXiv version is released on Sept 13, only few days before the ICLR submission deadline. Plus, till now the authors have not released their codes nor their finetuned weights.
>
> - Adjoint matching is proposed for flow matching, although some very special cases of diffusion models with appropriate noise schedule satisfy the so-called "memoryless" property, to which adjoint matching happens to be applicable if we may treat the trained diffusion model as a continuous process (i.e., a SDE). Our method, in comparison, can be applied to any MDP constructed by a diffusion model (for instance, we show that our method works with the MDP constructed with SDE-DPM-solver++ in Figure 18 and 19). What's more, our $\nabla$-DB objective can work for many other MDPs as it is derived under a non-diffusion-specific framework.
>
> We totally agree that it is definitely worth seeing the comparison between our method and adjoint matching despite their different applicability, and therefore we tried our best to implement adjoint matching by ourselves (as it is not open-sourced yet), do the experiments that are possible in this short period of time, and show the results in these anonymous links:
>
> - Comparison on reward convergence: [https://drive.google.com/file/d/1JpvOoKhAxni5-Z5NadhmzdDDQWEClZh8/view?usp=sharing](https://drive.google.com/file/d/1JpvOoKhAxni5-Z5NadhmzdDDQWEClZh8/view?usp=sharing)
> - Comparison on diversity evolution: [https://drive.google.com/file/d/1PYpn1h_O5_7APGRkucJFZmHtjpOJnyiV/view?usp=sharing](https://drive.google.com/file/d/1PYpn1h_O5_7APGRkucJFZmHtjpOJnyiV/view?usp=sharing)
> - Comparison on FID evolution: [https://drive.google.com/file/d/1Zb0kW_ySMm1b5aqkqFu0jADWXUMO5pne/view?usp=sharing](https://drive.google.com/file/d/1Zb0kW_ySMm1b5aqkqFu0jADWXUMO5pne/view?usp=sharing)
> - Comparion on generated images: [https://drive.google.com/file/d/1H4ulhKo4MKrV04bYKWTN4lfmebIofsZP/view?usp=sharing](https://drive.google.com/file/d/1H4ulhKo4MKrV04bYKWTN4lfmebIofsZP/view?usp=sharing)
>
> Both the quantitative and the qualitative results show that adjoint matching, compared to our method, generates images with less diversity and less prior preservation.
>
> ---
> We thank the reviewer in advance for the reviewer's time in reading this response, and sincerely hope that the reviewer may take our new experiments and explanations into consideration.

---

> > ### Comment · Reviewer_5N8Z · 2024-12-01
> >
> > I thank the authors for their effort in implementing adjoint matching and comparison to elegant.
> >
> > These results seem to show that there is a qualitative benefit for $\nabla$-DB method in comparison to adjoint matching. One concern that I have is that the definition of FID score. As mentioned in the manuscript, this is computed by the FID score "between images generated from the pre-trained model and from the fine-tuned model and take the average FID score over all evaluation prompts". This is not the traditional use of FID score and I think that this should be renamed to reflect this deviation from the norm.
> >
> > I think there should be some mention of limitations to this method. As mentioned in another review, this only works with SDE diffusion models, not with flow matching which has become used more frequently. Although the authors mentioned that this is not a significant limitation, I think that this is quite important. I hope that the authors can make further modifications to method in this work to show that this is possible for flow matching.

---

> ### Author Response · Authors · 2024-12-02
>
> We appreciate the reviewer's constructive comments.
>
> We agree with the reviewer that the term FID is a bit misleading and we will term it instead as "prompt-averaged FID" (this practice follows the way that the adjoint matching paper does with diversity scores).
>
> Regarding limitations: we will mention the limitations in our next draft, as we agree that flow matching is one of the mainstream models these days. In the meantime, we would like to point out that (which we are inspired by the reviewer to check), since the typical flow matching models like SD3 are those with *Gaussian conditional probability paths*, we have a very simple correspondence between the learned vector field $v(x,t)$ and the score function $\nabla_{x_t} \log p_t(x_t)$. For instance, for linear flow:
>
> $$\nabla_{x_t} \log p_t(x_t) = \frac{t-1}{t}v(x_t, t) - \frac{x}{t}$$
>
> Such a relationship indeed induces a corresponding SDE in this flow matching setting. As a result, we are able to use SD3 as an initialization and use $\nabla$-GFlowNet to obtained a finetuned diffusion model out of it. This is slightly different from directly obtaining a flow matching model, but arguably we may always compute the probability flow ODE or use DDIM-like solvers if ODE-style inference is preferred.
>
> Due to the extremely limited amount of time, we are less confident to show the reviewer empirical results before the end of the rebuttal period. Nevertheless, we are trying our best to produce the results, and at the same time hope our explanations above can alleviate some of your concerns.

---

### Official Review · Reviewer_y62f · 2024-11-03

**Soundness:** 2
**Presentation:** 2
**Contribution:** 3
**Rating:** 6
**Confidence:** 2

**Summary:**

The authors propose Nabla-GFlowNet (∇-GFlowNet) to efficiently finetune pretrained diffusion models. This approach addresses issues of limited sample diversity and slow convergence in existing methods by leveraging reward gradients through ∇-DB and its variant, residual ∇-DB. Empirical results show that residual ∇-DB enables fast, diversity-preserving finetuning of models like StableDiffusion on various realistic reward functions.

**Strengths:**

- The paper offers a comprehensive theoretical deduction of the proposed method, thoroughly explaining how the objectives nabla-DB and residual nabla-DB are derived.
- By introducing residual ∇-DB, the authors extend the applicability of their work to pretrained large-scale models, which is crucial.
- The paper enhances the quantitative evaluation of diversity in generated samples. By employing a broader range of metrics and more extensive comparisons.

**Weaknesses:**

- The current experimental setting appears somewhat outdated. To enhance the study's relevance, please consider using more recent schedulers and pre-trained models instead of DDPM or Stable Diffusion 1.5.
- The qualitative results shown in Figure 2 are confusing. Additional explanation is needed to clearly demonstrate the superiority of ∇-DB, as DDPO and DAG-DB also exhibit strong performance.
- A user study would be helpful for evaluating diversity.

**Questions:**

Please see weaknesses.

---

> ### Author Response · Authors · 2024-11-21
>
> We thank the review for the valuable comments and suggestions. We have made some further refinements on the paper presentation and the experiments (detailed in the general response).
>
> > The current experimental setting appears somewhat outdated. To enhance the study's relevance, please consider using more recent schedulers and pre-trained models instead of DDPM or Stable Diffusion 1.5.
>
> Thank you for raising this concern. We would like to first state that our paper follows the common practice in evaluating reward finetuning methods, for instance [1,2,3], almost all of which finetune with DDPM on StableDiffusion v1.5.
>
> That’s said, it is definitely interesting to see how our method may generalize to some other MDPs constructed by different diffusion SDE solvers. For this purpose, we constructed another MDP with the schedule of SDE-DPM-Solver++ [4] and showed in the appendix (Figure 18 and 19) that our method still works well in this setting.
>
> Regarding pretrained models, SD v1.5 is probably the best open-sourced large diffusion models for verifying different methods. SDXL can be too large and require more compute that a typical academic lab does not afford. SD3 is a flow matching model (despite the word “diffusion” in the name “StableDiffusion”), of which the sampling process is deterministic (given an initial Gaussian noise) by solving an ordinary differential equation (ODE). As GFlowNets are probabilistic models, they by design do not model deterministic processes and therefore we do not consider SD3 as a suitable model to benchmark methods for diffusion finetuning.
>
> And we would also like to share our insight that, since our method is mostly algorithm- and model-agnostic (as it is indeed derived from a general probabilistic model perspective) for diffusion models, we confidently believe that there is not much gap between the results on different pretrained diffusion models and different MDP.
>
> > The qualitative results shown in Figure 2 are confusing. Additional explanation is needed to clearly demonstrate the superiority of ∇-DB, as DDPO and DAG-DB also exhibit strong performance.
>
> We apologize for our unclear presentation on this result. We have revised the figure and the captions.
>
> In the revised draft (Table 1 and Fig 12 & 13), our method achieves the best performance (in the sense that it has comparable speed to ReFL while maintaining good diversity & prior preservation) with our latest modifications:
>
> - Introduction of attenuating scaling on the predicted rewards (so that the less reliable predicted rewards at time steps far from the generated samples are down-weighted)
> - Fix on the numerical issues in mixed-precision training
>
> > A user study would be helpful for evaluating diversity.
>
> We appreciate the reviewer’s suggestion —- we are currently working on it and hopefully we can report the results very soon.
>
> ---
> [1] Training Diffusion Models with Reinforcement Learning. Kevin Black, Michael Janner, Yilun Du, Ilya Kostrikov, Sergey Levine. ICLR 2024
>
> [2] Directly Fine-Tuning Diffusion Models on Differentiable Rewards. Kevin Clark, Paul Vicol, Kevin Swersky, David J. Fleet. ICLR 2024
>
> [3] Improving GFlowNets for Text-to-Image Diffusion Alignment. Dinghuai Zhang, Yizhe Zhang, Jiatao Gu, Ruixiang Zhang, Josh Susskind, Navdeep Jaitly, Shuangfei Zhai. https://arxiv.org/abs/2406.00633
>
> [4] DPM-Solver++: Fast Solver for Guided Sampling of Diffusion Probabilistic Models. Cheng Lu, Yuhao Zhou, Fan Bao, Jianfei Chen, Chongxuan Li, Jun Zhu. ICLR 2023

---

> > ### Author Response · Authors · 2024-11-23
> >
> > We would like to thank the reviewer for their patience. Here are the results of a simple user study, where we randomly picked 5 categories in animal prompts and for each generated 16 images to test on our models finetuned on Aesthetic Score with (residual $\nabla$-DB) and the baselines (Residual DB, DDPO, ReFL and DRaFT-LV). For each category, we present the corresponding images generated for all 5 methods and ask 3 questions:
> > 1. Which set is the most aesthetic one?
> > 2. Which set is the most diverse one?
> > 3. Only consider those images you feel aesthetic. Now, which one is the most diverse?
> >
> > We collected 27 responses and computed the win rate averaged on 5 prompts:
> >
> > | Method | Q1 | Q2 | Q3 |
> > |------------------|------------------|------------------|------------------|
> > |  Residual $\nabla$-DB | 63.7%   | 52.6%    | 61.7%    |
> > | Residual DB                 | 7.4%     | 0%         |  0%                        |
> > | DDPO                           | 11.9%   | 10.4%    |  8.9%                        |
> > | ReFL                             | 5.9%    | 18.5%    |  13.3%                       |
> > | DRaFT-LV                     | 11.1%    | 18.5%   |  17.0%                        |
> >
> >
> > The results of this user study are largely consistent with our quantitative results with the ground truth reward function (Aesthetic Score) and the proxy diversity score (DreamSim diversity).

---

> ### Comment · Reviewer_y62f · 2024-11-26
>
> Thanks for your reply. The authors address most of my concerns. I decide to maintain my initial rating 6.

---

> > ### Author Response · Authors · 2024-12-03
> >
> > As it is coming to the end of the rebuttal period, we would like to thank the reviewer for their time and efforts and their continued overall positive rating, and we are glad that we addressed your concerns.

---

### Official Review · Reviewer_BEKY · 2024-11-04

**Soundness:** 3
**Presentation:** 3
**Contribution:** 4
**Rating:** 6
**Confidence:** 3

**Summary:**

This paper introduces an interesting method to address the challenges of fine-tuning multistep sampling in diffusion models. It employs GFlowNets to incorporate a middle term, $F(x_t)$ or $g_\phi(x_t)$, which allows the reward score to effectively influence different timesteps.  This method successfully eliminates the need to train a reward model that accepts noisy input. This paper implements their idea in both theoretical and practical contexts. Section 3.1 covers the theoretical aspect, while sections 3.2 and 3.3 address the practical application. Experiments also show that this method can enhance reward tuning in diffusion models.

**Strengths:**

1. This paper presents a new method for addressing the challenges of fine-tuning multistep sampling in diffusion models using GFlowNets. This method effectively eliminates the need to train a reward model that processes noisy input.
2. This paper implements their idea in both theoretical and practical contexts. Section 3.1 covers the theoretical aspect, while sections 3.2 and 3.3 address the practical application.

**Weaknesses:**

The main weakness is in the experiment part.
1. The function $g_\phi(x_t)$ is an interesting and reasonable choice for achieving the fitness task; however, it results in approximately zero vectors, with a terminal constraint of $g_\phi(x_T) = 0$. It remains unclear whether Unet is a suitable option for this purpose.
2. The regularization term appears significant, with $\lambda=1000$ in the Aesthetic Score experiments and $\lambda=100$ in the HPSv2 experiments. However, Section 3.2 states that it "may eventually over-optimize the reward and thus neglect the pretrained prior." Is Section 3.2 more helpful for preventing over-optimization than the regularization term?
3. The method is interesting and useful, but the paper's claim of "Diversity-Preserving" in the title creates a gap. Does this imply that the method theoretically ensures better diversity, or is it based solely on observational results? From my perspective, this method prioritizes improving information backpropagation for fine-tuning multistep sampling rather than enhancing diversity.

If the authors can resolve my issue, I will contemplate raising my score.

**Questions:**

See the weaknesses.

---

> ### Author Response · Authors · 2024-11-21
>
> We appreciate the review's acknowledge on our contribution and their valuable comments. We have made some further refinements on the paper presentation and the experiments (detailed in the general response).
>
> > The function $g_\phi(x_t)$ is an interesting and reasonable choice for achieving the fitness task; however, it results in approximately zero vectors, with a terminal constraint of $g_\phi(x_T)=0$. It remains unclear whether Unet is a suitable option for this purpose.
>
> Good point! While the $g_\phi(x)$ function at the terminal state is zero, in general it is moderately far from zero for states far from the terminal state. Indeed, this is the reason why we resort to GFlowNet to solve this issue. We therefore argue that it is necessary to predict a vector field from the input, for which U-Net is a good choice. Indeed, the results with $w_B = 0$ in our paper shows that when the detailed balance constraints are not well obeyed, the model suffers from worse diversity and worse prior following, which partially demonstrates the importance of learning a good $g_\phi(x_t)$.
>
> > The regularization term appears significant, with $\lambda=1000$ in the Aesthetic Score experiments and $\lambda=100$ in the HPSv2 experiments. However, Section 3.2 states that it "may eventually over-optimize the reward and thus neglect the pretrained prior." Is Section 3.2 more helpful for preventing over-optimization than the regularization term?
>
> The regularization is indeed more concerned about training stability, as the training examples are generated by the policy currently being finetuned (i.e., the so-called on-policy training in reinforcement learning literature). RL Algorithms like TRPO [1] and PPO [2] have similar regularization to avoid sudden collapse of the policy.
>
> We stated the overfitting issue because we set a relatively high reward-to-prior ratio (i.e., the $\beta$ parameter) for faster convergence. As a result, if we train sufficiently long (say several days), it is possible for the policy to completely ignore the prior.
>
> > The method is interesting and useful, but the paper's claim of "Diversity-Preserving" in the title creates a gap. Does this imply that the method theoretically ensures better diversity, or is it based solely on observational results? From my perspective, this method prioritizes improving information backpropagation for fine-tuning multistep sampling rather than enhancing diversity.
>
> Thanks for raising your concern and we apologize for our unclear previous descriptions.
>
> By “diversity-preserving”, we mean that our method aims to **sample** from the target probability distribution, versus to **maximize** the probability distribution (as in DDPO and ReFL). Here we give a simple example: suppose that the policy is a single 1D Gaussian distribution (i.e., running diffusion for only one step) with both the mean and variance parameters learnable, and the target distribution is a mixture of two Gaussians of different variance and moderately far from each other. With the **reward maximization** objective, the optimal solution is to have the policy to always (i.e., deterministically) output the point with the maximum probability (i.e., a Dirac delta distribution), while with the **sampling objective** one aims to minimize the distribution distance between the policy and the target probability, which in general leads to non-zero overlap between distributions in the visualization of both distributions.
>
> Therefore, we argue that these non-sampling baseline methods (DDPO, ReFL and DRaFT) are more prone to mode collapse (since if they find a mode in the distribution, there is less incentive for these models to discover other modes).
>
> ---
> [1] Trust Region Policy Optimization. John Schulman, Sergey Levine, Philipp Moritz, Michael I. Jordan, Pieter Abbeel. ICML 2015
>
> [2] Proximal Policy Optimization Algorithms. John Schulman, Filip Wolski, Prafulla Dhariwal, Alec Radford, Oleg Klimov. https://arxiv.org/abs/1707.06347

---

> > ### Author Response · Authors · 2024-11-30
> >
> > Dear Reviewer BEKY,
> >
> > We wanted to follow up and check if you have any concerns or questions regarding our previous response. If so, we would be more than happy to provide further clarification or address them in detail. We would also like to share with you that we have included additional experiments in response to some other reviewers.
> >
> > We greatly appreciate your time and effort in reviewing our response and look forward to your feedback.
> >
> > Best,
> > Paper 8475 Authors

---

> > > ### Comment · Reviewer_BEKY · 2024-12-03
> > >
> > > The author partially addressed my concerns, and considering everything, I agree to accept this paper.
> > > 1. I suggest that the author should include these points in the latest version of the paper, especially regarding diversity, as it was mentioned in the title.
> > > 2. I have an additional question; given the deadline, the author does not need to respond but I recommend considering it for future work. I'd like to ask if a good reward could completely eliminate the need for a regularization term or whether the existence of a regularization term is merely a patch for currently unexplained issues within rewards.

---

> ### Author Response · Authors · 2024-12-03
>
> We appreciate the reviewer's acknowledgement on our contribution and suggestions. We will make the diversity claim clearer in our next draft.
>
> **Regarding the regularization term.** The issue is rooted in the high *variance* nature of exploration in policy optimization (even though the objective is unbiased) plus the highly complex optimization landscape, especially when we are dealing with high-dimensional action spaces (in our case, the action is some denoised image). RL methods, which are very akin to our method, are shown in the literature of RL, control theory and optimization that the underlying optimization problem is hard. For instance, the famous maximization bias problem shows that even for simple unbiased reward functions in some tiny MDP of few states, a model can take a long time to overcome overestimations caused by early learning dynamics [1]. For another example: it is shown that the optimization with the so-called temporal difference (one of the core concepts in RL) can be unstable if certain conditions are violated [2].
>
> Empirically speaking, even in relatively small-scale control problems (compared to an action space of images), like the classical benchmark of Mujoco tasks (e.g., controlling a 27-degree-of-freedom humanoid to walk in simulated environments), naive RL algorithms hardly work well without regularization. There are papers to investigate the importance of regularization of RL algorithms, for example [3]. Due to the similarity between our method and RL ones, we expect regularization as an important component to alleviate the burden in tuning optimization hyperparamters.
>
> That's said, the field of deep RL has been developed for many years and therefore many standard and widely accepted and applicable engineering tricks besides KL regularization, just as techniques like gradient clipping, Xavier initialization (for ResNets), orthogonal initialization (for LSTM) and batch normalization in training deep neural nets for classification. Many of these RL techniques are used in large-scale systems like AlphaGo and AlphaStar. Therefore, we are relatively confident to claim that the use of RL regularization techniques in GFlowNet-related methods (including ours) is not a big issue.
>
> **Regarding picking reward functions.** As we aim to accommodate different reward functions (*e.g.*, a large transformer-based neural reward model learned from noisy human preference data, or rewards from accurate physical simulators), we do not make specific assumptions on what the rewards are, so as to find methods that are generalizable to different reward functions.
>
> We hope our explanations may resolve some of your concerns.
>
> ---
>
> [1] https://web.stanford.edu/class/cs234/CS234Win2023/slides/lecture6post.pdf (Slides from the RL course by Emma Brunskill)
>
> [2] Simplifying Deep Temporal Difference Learning. ICLR 2025 Conference Submission7833. https://openreview.net/forum?id=7IzeL0kflu
>
> [3] Regularization Matters in Policy Optimization - An Empirical Study on Continuous Control. Zhuang Liu, Xuanlin Li, Bingyi Kang, Trevor Darrell. ICLR 2021. https://openreview.net/forum?id=yr1mzrH3IC

---

### Official Review · Reviewer_ZKLE · 2024-11-07

**Soundness:** 3
**Presentation:** 3
**Contribution:** 2
**Rating:** 6
**Confidence:** 4

**Summary:**

The paper addresses the problem of fine-tuning the pretrained diffusion models on a target reward while aiming for 1) preserving diversity of generated images and 2) fast convergence. It proposes Nabla-GFlowNet to do so, inspired by generative Flow Nets (GFlow-Nets) that sample with unnormalized density of the reward function. Experiments on different benchmarks show that the proposed method generally achieves the best diversity vs reward trade-off compared to baselines.

**Strengths:**

- The proposed idea is based on the generative flow nets, which makes it intuitive and straightforward.
- The Nabla-GFlowNet can leverage the first order information of the reward function (gradient) while the baselines only use the zero-order information.
- The experimental results show that the proposed method can generally achieve the best diversity vs. reward trade-off frontiers.

**Weaknesses:**

- I think the "predicted reward" estimation in Eq. 15 can be severely unreliable, especially for the high-noise time-steps of the diffusion model. The predicted clean image will be noisy, and if the reward function is calculated by a model that has been trained on not noisy images, the predicted reward will be inaccurate.

- The parameter \lambda and the output regularization described in Page 7 seems to be crucial to the model's performance, but they are not the paper's contribution.

- The qualitative samples that are compared with the baselines are only for the aesthetic score. I think a qualitative comparison on the HPSv2 can be more valuable and insightful about the model's performance.

**Questions:**

- I suggest that the authors include the qualitative comparative results for the HPS-v2 reward in the paper.

If the authors address my concerns, I am willing to increase my score.

---

> ### Author Response · Authors · 2024-11-21
>
> We appreciate the reviewer for acknowledging our contribution and raising their concerns.
>
> > I think the "predicted reward" estimation in Eq. 15 can be severely unreliable, especially for the high-noise time-steps of the diffusion model.
>
> Thanks for your insightful comment. It is undoubtedly true that the predicted reward is never accurate. Indeed, this is the core reason why we need to learn the residual flow function to correct this error, which differentiates our method from others like ReFL (which also computes the predicted reward).
>
> By design, our method does not rely on the predicted reward — it is only a technique to set a good initialization for the flow function. If we are allowed to parameterize the flow function with a sufficiently large neural net and to spend much more compute (e.g., very large batch size) to optimize the flow function, we do not need this predicted reward because the detailed-balance (DB) consistency losses will propagate the reward signal on the terminal state back to all previous ones.
>
> In practice, as one of our objectives is to achieve fast convergence for reward finetuning, we are not allowed to update the policy for too many steps with small learning rate / very large batch size. And indeed as what you may have been concerned about, the inaccurate reward prediction, under high learning rate, can lead to worse performance. Both to answer your question and to address this issue, we experimented to attenuate the scale of the predicted reward according to the diffusion time step — the further away the diffusion time step is from the data distribution, the less weight we place on the predicted reward. We found that not only the performance (in terms of reward vs diversity trade-off) significantly improves with convergence speed slightly slower. To better demonstrate the effectiveness of our method, in the main text we show the results of our method with reward scaling, but show the comparison between w/ and w/o attenuation in Figure 20 and 21 in the appendix.
>
> > The parameter $\lambda$ and the output regularization described in Page 7 seems to be crucial to the model's performance, but they are not the paper's contribution.
>
> The regularization term is a common technique in the literature of reinforcement learning (especially for on-policy settings where the training trajectories are sampled from the policy currently being finetuned) to avoid too abrupt change in the policy during training, as it may lead to collapse of training. While it may be more common to find papers using KL divergence to do this (for instance, in the very popular methods of TRPO [1] and PPO [2]), we use Fisher divergence (therefore the L2 loss between diffusion policy outputs —- the gradients of the log probabilities). Indeed, the regularization term is used in one of our baselines DAG-DB [3]. Empirically, as long as this divergence is sufficiently large to prevent divergence in training loss, the results will be good.
>
> > I think a qualitative comparison on the HPSv2 can be more valuable and insightful about the model's performance.
>
> We thank the reviewer for pointing this out. We have included more qualitative results on HPSv2 (with improved performance due to our modifications) in Figure 4 in the main text and Figure 25, 26 in the appendix. In addition, we now include the results on another reward function, ImageReward [4], to demonstrate the capability (Figure 5).
>
> ---
>
> [1] Trust Region Policy Optimization. John Schulman, Sergey Levine, Philipp Moritz, Michael I. Jordan, Pieter Abbeel. ICML 2015
>
> [2] Proximal Policy Optimization Algorithms. John Schulman, Filip Wolski, Prafulla Dhariwal, Alec Radford, Oleg Klimov. https://arxiv.org/abs/1707.06347
>
> [3] Improving GFlowNets for Text-to-Image Diffusion Alignment. Dinghuai Zhang, Yizhe Zhang, Jiatao Gu, Ruixiang Zhang, Josh Susskind, Navdeep Jaitly, Shuangfei Zhai. https://arxiv.org/abs/2406.00633
>
> [4] ImageReward: Learning and Evaluating Human Preferences for Text-to-Image Generation. Jiazheng Xu, Xiao Liu, Yuchen Wu, Yuxuan Tong, Qinkai Li, Ming Ding, Jie Tang, Yuxiao Dong. NeurIPS 2023

---

> > ### Comment · Reviewer_ZKLE · 2024-11-24
> > **Response to Authors' Rebuttal**
> >
> > I thank the authors for their efforts for the rebuttal. The rebuttal addressed most of my concerns, and I raise my score to 6. Specially, I appreciate the experiments when attenuating the scale of reward value for noisier time-steps, and I recommend that the authors include the weighting scheme that they used for this experiment in the supplementary. Yet, I didn't give a higher score because I believe the novelty and technical contribution of the paper is relatively limited.

---

> ### Author Response · Authors · 2024-11-25
>
> We sincerely appreciate the reviewer's time and efforts in reviewing our response, and their further acknowledgement on our contribution. We are glad that we addressed your concerns.
>
> As suggested by the reviewer, we have included the weighting scheme at the end of Section 3 in the main text.
>
> In the meantime, we would also like to take this opportunity to further clarify and highlight the novelty and technical contributions of our work from two perspectives:
>
> **From the perspective of reward finetuning.**
>
> Our proposed method is a *practical, scalable and provable* method that
>
> - works and scales well on **large diffusion models**;
> - has **theoretical guarantees** of unbiased optimal solution to the target distribution; and,
> - achieves **Pareto improvements** on reward convergence and diversity & prior preservation on **all reward functions** we experimented with.
>
> While the derivation naturally follows GFlowNet principles, we believe that not only the novel combination of these principles, with the scalability to large models, is a significant step. Plus, we believe the introduction and application of this new GFlowNet method to reward finetuning of diffusion models builds significant, and perhaps previously overlooked, connections between fields.
>
> **From the perspective of GFlowNets.**
>
> - We propose the **first** GFlowNet objective that leverages **gradient signals** in training GFlowNets, while all other GFlowNet methods only use zeroth-order signals.
> - Our method demonstrates scalability to large diffusion models, such as StableDiffusion with a substantial number of sampling steps—something that few existing GFlowNet approaches have achieved.
>
> We believe these aspects constitute a novel contribution to the GFlowNet literature, which we regret not having fully explained in our paper or earlier responses. We sincerely hope this perspective provides further context for evaluating our work.

---

### Author Response · Authors · 2024-11-21
**General response to the reviewers**

We sincerely thank the reviewers for their time and effort in reviewing our paper and making valuable suggestions. We have uploaded a revised draft of our paper. The major differences are:

- We have 1) fixed the numerical issue during training, 2) introduced **time-dependent weighting on the predicted rewards** of the residual $\nabla$-DB loss. As a result, our proposed method is now able to achieves a **much better trade-off between reward, diversity and prior-following on all reward models**, shown in Table 1 and Figure 6, 7, 12 & 13 for the main experiments in the revised draft. We have therefore also revised our Figure 2 on the results using Aesthetic Score.
- To further demonstrate the effectiveness of our methods and in response to Reviewer zWHc and 5qSt, we include more qualitative results in Figure 3, 4, 5 as well as the last three pages in the appendix.
- We now introduce a **new hyperparameter to control the strength of the pretrained prior**, for which the ablation results are shown in Figure Figure 16 and 17.
- We now include a **new metric of prior preservation** by measuring the FID score (in general, lower the better) between samples generated by the finetuned model and the pretrained model (as our objective is $P_\text{pretrained}(x)^\eta * R(x)^\beta$). For a fair comparison, we draw the Pareto frontier to show the FID scores of models at roughly the same reward. Figure 7 shows that our method is significantly better in this dimension.
- We now include **another reward function ImageReward** [1], a general-purpose text-to-image human preference reward model.
- In response to Reviewer y26f and 5qSt, we include some more details in the experiment section to avoid confusion.
- In Figure 2, we used to show the final results (trained with 200 update steps) of all methods. While it is indeed the case that our method is more robust, it might be a bit unfair as one may perform early stopping when finetuning models. Therefore, we now visualize the samples from the best model checkpoint (throughout the finetuning process) for each method with an average reward of the visualized set of images, while we qualitatively show the training collapse issue of the baseline methods in Figure 3.
- Due to the limited space in the paper, we moved some figures like the one for ablation study on trajectory subsampling rate to the appendix.
- In response to Reviewer y62f, we have experimented with a different scheduler. Specifically, we pick **SDE-DPM-Solver++** and construct an MDP. In Figure 18 and 19, we show that our method is still able to perform well.
- For every experiment we now show the standard deviation of metrics with 3 random seeds.
- We polished the figures, the captions the related work section for better flow.

---

### Author Response · Authors · 2024-11-24
**Following-up on reviewers' concerns**

Dear Reviewers and AC,

We thank again for your time and efforts in reviewing our paper and going through the possibly long responses to your questions. As it is coming to the end of the rebuttal period, we would like to learn if we have addressed your concerns or the reviewers have any other questions. If any, we are more than happy to answer them.

Many Thanks,

Paper 8475 Authors

---

### Author Response · Authors · 2024-11-25
**Summary of our contributions**

We would like to copy and paste parts of our supplementary response to Reviewer ZKLE, to give a better summarization and explanation on our contributions (especially from the perspective of GFlowNets):

**From the perspective of reward finetuning.**

Our proposed method is a *practical, scalable and provable* method that

- works and scales well on **large diffusion models**;
- has **theoretical guarantees** of unbiased optimal solution to the target distribution; and,
- achieves **Pareto improvements** on reward convergence and diversity & prior preservation on **all reward functions** we experimented with.

**From the perspective of GFlowNets.**

- We propose the **first** GFlowNet objective that leverages **gradient signals** in training GFlowNets, while all other GFlowNet methods only use zeroth-order signals.
- Our method demonstrates scalability to large diffusion models, such as StableDiffusion with a substantial number of sampling steps—something that few existing GFlowNet approaches have achieved.

---

### Meta-Review · Area_Chair_q9LY · 2024-12-20

**Metareview:**

This paper proposes a novel GFlowNet method dubbed Nabla-GFlowNet (abbreviated as $\nabla$-GFlowNet), together with an objective called $\nabla$-DB, plus its variant residual $\nabla$-DB for finetuning pretrained diffusion models, to achieve fast yet diverse sampling.

All 5 reviewers give a final rating of 6. I am on board with them and recommend accept.

**Additional Comments On Reviewer Discussion:**

Initial concerns include: the reliability of the "predicted reward" estimation in Eq. 15; the claim of diversity preserving is untenable; the current experimental setting appears somewhat outdated; this paper may be optimizing for a bias distribution while preserving diversity, etc.

Most of these concerns have been well addressed after the rebuttal.

---

### Decision · Program_Chairs · 2025-01-22

Accept (Poster)